# Carbon redirection via tunable Fenton-like reactions under nanoconfinement toward sustainable water treatment

Xiang Gao[1,4], Zhichao Yang[1,2,4], Wen Zhang [ID][3] & Bingcai Pan [ID][1,2] ✉

The ongoing pattern shift in water treatment from pollution control to energy recovery challenges the energy-intensive chemical oxidation processes that have been developed for over a century. Redirecting the pathways of carbon evolution from molecular fragmentation to polymerization is critical for energy harvesting during chemical oxidation, yet the regulation means remain to be exploited. Herein, by confining the widely-studied oxidation system—$Mn_3O_4$ catalytic activation of peroxymonosulfate—inside amorphous carbon nanotubes (ACNTs), we demonstrate that the pathways of contaminant conversion can be readily modulated by spatial nanoconfinement. Reducing the pore size of ACNTs from 120 to 20 nm monotonously improves the pathway selectivity toward oligomers, with the yield one order of magnitude higher under 20-nm nanoconfinement than in bulk. The interactions of $Mn_3O_4$ with ACNTs, reactant enrichment, and pH lowering under nanoconfinement are evidenced to collectively account for the enhanced selectivity toward polymerization. This work provides an adaptive paradigm for carbon redirection in a variety of catalytic oxidation processes toward energy harvesting and sustainable water purification.

The aggravated water pollution and water scarcity pose global threats to the sustainable development of the economic society and ecosystems[1,2], propelling the innovation of solutions for water purification and reuse. Chemical oxidation stands as one of the exemplary measures that has been contributing incomparably to vanquishing waterborne diseases and water pollution since the early 1900s[3,4]. Aiming to eliminate contaminants to meet stringent effluent standards, the development of chemical oxidation processes has long been centered on improving degradation efficiency. However, the substantial energy embedded in wastewater (-13–28 kJ gCOD$^{-1}$; COD represents chemical oxygen demand)[5], in sharp contrast to the considerable energy required for treatment (-3% of the global electricity)[6,7], is driving a paradigm shift in water treatment from pollution control to energy recovery and lower carbon footprint[8], posing a significant challenge to the traditional energy-intensive oxidation processes. A new paradigm of chemical oxidation that redirects pathways of carbon evolution toward energy harvesting while controlling pollution is therefore urgently desired.

In a typical chemical oxidation process, the chemical bonds of contaminants are cleaved with the assistance of external energy, producing smaller molecules and/or $CO_2$ while releasing energy simultaneously into the water in an unorganized manner. Conceivably, one potential strategy for energy harvesting of dissolved organic contaminants is to convert them into polymers of higher molecular weights, which are usually of low solubility and could be separated from water by adsorption[9,10], filtration[11,12], or flotation[13] for potential chemical refining[14–17] and electricity production[12,18,19]. The desirable polymerization does not require the decomposition of the carbon skeletons of contaminants, which not only reduces the input of chemicals or energy but also preserves the embedded chemical energy.

[1]State Key Laboratory of Pollution Control and Resources Reuse, Nanjing University, Nanjing, China. [2]Research Center for Environmental Nanotechnology (ReCENT), School of Environment, Nanjing University, Nanjing, China. [3]John A. Reif, Jr. Department of Civil and Environmental Engineering, New Jersey Institute of Technology, Newark, NJ, USA. [4]These authors contributed equally: Xiang Gao, Zhichao Yang. ✉e-mail: bcpan@nju.edu.cn

For instance, the conversion of phenol (PhOH) to $CO_2$ in the traditional oxidation system consumes at least seven equivalents of $O_2$ and releases ~3.1 MJ mol$^{-1}$ of energy into the water, whereas polymerization of PhOH into tetramers with an internal energy of ~11.7 MJ mol$^{-1}$ theoretically necessitates no more than three equivalents of $O_2$. Despite a recent flurry of activity to revisit a variety of oxidation systems for the possibility of polymerization, very limited successful cases[9–12,17,18,20] prompt more efforts in exploring how to direct oxidation pathways from degradation to polymerization.

In nature, enzyme-mediated synthesis of lignin in plants proceeds via polymerization of phenolic compounds[21]. The involved enzymes also take sophisticated control over the oxygenation reactions with exclusive selectivity in water[22–25]. In addition to the functional active sites for binding and catalysis, the unique microenvironment exerts pivotal nanoconfinement effects to favor oxidative coupling both kinetically and thermodynamically[23,26–29]. Mimicking the nanoconfinement effects in enzymes also serves as a useful tool for engineering polymer synthesis[30,31]. The physicochemical properties of the supports, particularly the pore diameter, would influence the electronic structure of catalytic sites, diffusion of reactants/products, as well as the orientation and effective concentration of reactants to alter the reaction outcomes[30,32,33]. With respect to water treatment, nanoconfinement generally enhances the activity and robustness of catalytic oxidation processes[34–36], yet the potential for steering the oxidation pathways remains to be exploited.

Here, we take a step to bridge the knowledge gap on how the spatial size of nanoconfinement redirects carbon evolution pathways in chemical oxidation-based water treatment. A typical Fenton-like oxidation process in bulk[9,37,38], i.e., $Mn_3O_4$ catalytic activation of peroxymonosulfate (PMS), was confined in amorphous carbon nanotubes (ACNTs) with varied diameters (i.e., 20 nm, 55 nm, and 120 nm, respectively). PhOH, a representative aromatic contaminant ubiquitous in water, features a simple structure and was used as the model target to facilitate product identification. We methodically examined the effect of nanoconfinement on product distribution under various conditions and elucidated the underlying mechanisms. In particular, we unveil unequivocally a spatial size-driven selectivity of oxidation products from fragmented molecules to polymers, as a result of the synergistic contributions from the host-guest interactions, reactant enrichment, and the solution pH lowering under nanoconfinement.

## Results and discussion
### Structure of the catalysts

The scheme of catalyst fabrication is shown in Fig. 1a. Transmission electron microscopy (TEM) images show that ACNTs, templated by the anodic aluminum oxide (AAO) membranes with varied pore sizes, feature a straight nanotubular structure with similar wall thickness (7.2 ± 0.4 nm) but different pore diameters of 19.8 ± 2.4 nm, 55.4 ± 3.5 nm, and 121.4 ± 4.6 nm, respectively (Supplementary Fig. 1a). A closer observation of the walls reveals the absence of concentric scrolled graphitic multilayers (Supplementary Fig. 1b) indicating the amorphous characteristics of all three ACNTs. This observation aligns well with the broad diffraction peaks at 26° in the X-ray diffraction (XRD) spectra (Supplementary Fig. 2). The $Mn_3O_4$ nanoparticles (NPs)

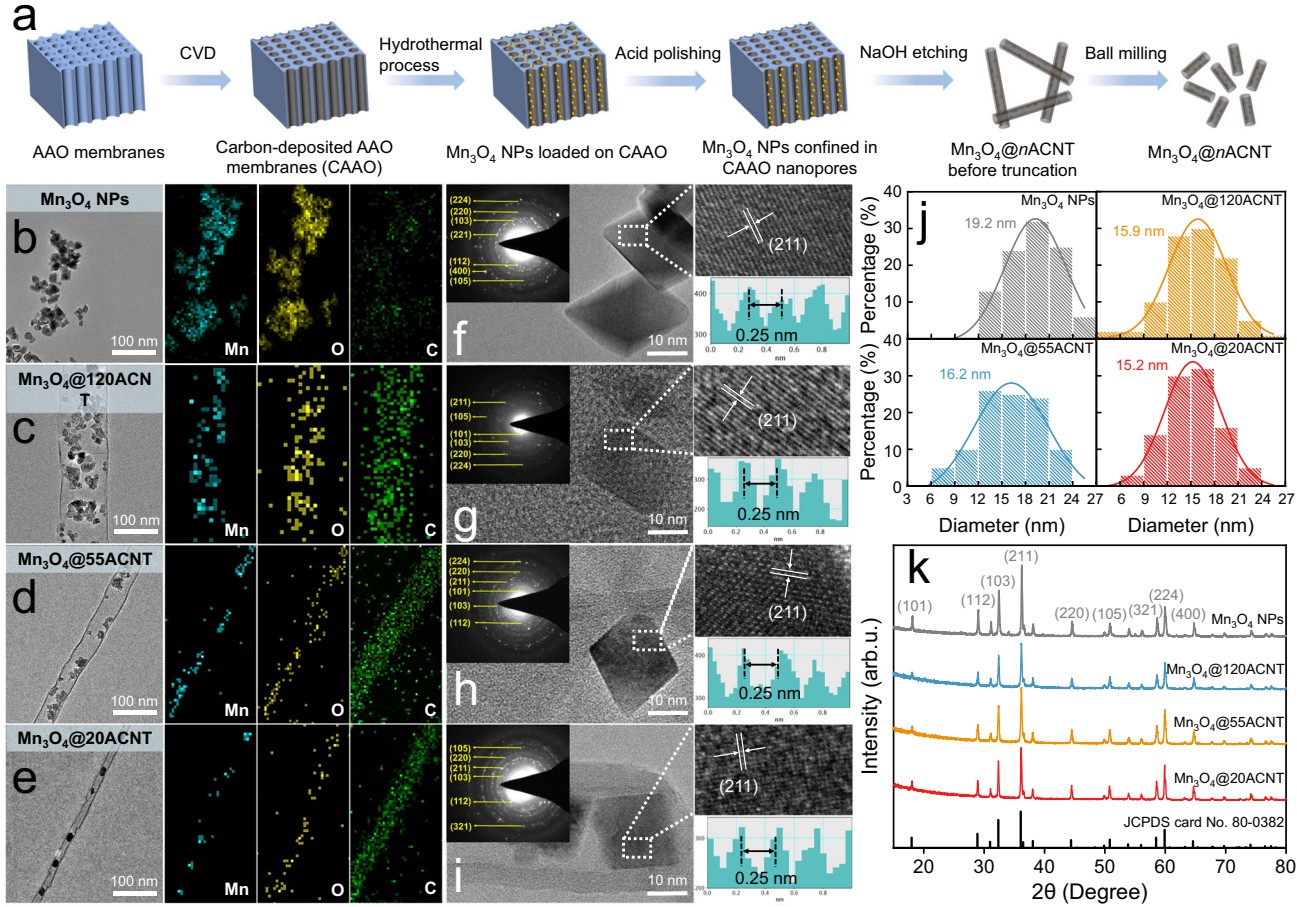

**Fig. 1 | Structural characterization of the nanocomposite catalysts. a** Scheme of catalyst fabrication. **b**–**e** TEM images and EDX elemental mappings of bulk $Mn_3O_4$ NPs and $Mn_3O_4@n$ACNT. Representative HRTEM images of the bulk $Mn_3O_4$ NPs (**f**), and those confined in 120ACNT (**g**), 55ACNT (**h**), and 20ACNT (**i**). **j** Size-distribution histograms of $Mn_3O_4$ NPs in different catalysts. **k** XRD patterns of the catalysts.

were encapsulated inside these ACNTs via hydrothermal reactions to produce the catalysts named Mn$_3$O$_4$@$n$ACNT ($n$ = 20, 55, and 120, representing the pore diameters of ACNTs in nm). The tube lengths of Mn$_3$O$_4$@$n$ACNT were truncated to 1.0–2.0 μm to alleviate the possible transport resistance of reactants in nanopores (Supplementary Fig. 3). Thanks to the template method, the Mn$_3$O$_4$ NPs were all distributed randomly on the internal walls of ACNTs, as shown in TEM images coupled with energy-dispersive X-ray spectroscopy (EDX) elemental mapping (Fig. 1b–e). Confinement of Mn$_3$O$_4$ NPs inside ACNTs can also be reflected by the much weaker Mn 3 s peaks of Mn$_3$O$_4$@$n$ACNT compared to those of bulk Mn$_3$O$_4$ NPs due to the limited probe length of the photoelectrons (~5.0 nm, Supplementary Fig. 4). The Mn contents in the three catalysts were very close as determined by acid digestion (i.e., 19.4 wt% for Mn$_3$O$_4$@20ACNT, 20.6 wt% for Mn$_3$O$_4$@55ACNT, and 21.3 wt% for Mn$_3$O$_4$@120ACNT).

The confined Mn$_3$O$_4$ NPs and the bulk counterpart were extensively characterized by high-resolution TEM (HRTEM), XRD, and Raman analysis. HRTEM images indicate that Mn$_3$O$_4$ NPs in all the catalysts exhibit an octahedral rhombus shape with a dominant (211) crystal lattice plane (Fig. 1f–i). The sizes of Mn$_3$O$_4$ NPs are fairly uniform, falling in the range of 15.0–24.0 nm with comparable average sizes (15.2 ± 4.0 to 19.2 ± 6.3 nm, Fig. 1j). XRD patterns of the catalysts exhibit the characteristic diffraction peaks at 29.0°, 32.4°, 36.0°, and 60.0° (Fig. 1k), corresponding to Mn$_3$O$_4$ crystals with a spinel structure (JCPDS No. 80-0382). Moreover, the only discernible peaks at 654 cm$^{-1}$ in the Raman spectra of the catalysts can be assigned exclusively to the Mn–O A1g vibrational mode of Mn$_3$O$_4$ NPs (Supplementary Fig. 5). Together, the above results unambiguously illustrate that, apart from the diameter of nanopores, these catalysts are of satisfactory similarity in structure, which confers a rationality to unravel how spatial nanoconfinement affects the Fenton-like reaction.

## Tunable PhOH conversion under nanoconfinement

The conversion of PhOH in both confined Mn$_3$O$_4$@$n$ACNT/PMS and bulk Mn$_3$O$_4$/PMS systems was evaluated and compared under similar conditions. In the absence of either PMS or catalyst, negligible conversion of PhOH was observed (Supplementary Fig. 6). Bulk Mn$_3$O$_4$ NPs exhibited moderate reactivity for PMS activation and ~42% of PhOH was oxidized within 150 min, experiencing a lag phase of 60 min followed by a relatively rapid phase. By confining Mn$_3$O$_4$ NPs inside ACNTs, the conversion of PhOH was significantly improved to 90–100% (Fig. 2a). In addition, the lag phase of PhOH oxidation was shortened and even disappeared in Mn$_3$O$_4$@20ACNT/PMS. Note that ACNTs were inert toward PMS activation due to the poor crystalline structure (Supplementary Fig. 7). The slight leaching of Mn during the oxidation processes (<20 μM) negligibly contributed to PMS activation regardless of the presence of ACNTs (Supplementary Fig. 8). These findings indicate that Mn$_3$O$_4$ NPs, whether confined or not, dominate PMS activation for efficient PhOH conversion. The accelerated kinetics of PhOH conversion under nanoconfinement did not correlate with the adsorption equilibrium of PMS or PhOH on the catalysts (Supplementary Figs. 9 and 10). It is reminiscent of the progressively intensified PhOH oxidation in the classical Fe(II)/H$_2$O$_2$ system[39], where the accumulated benzoquinone (BQ)/hydroquinone (HQ) serves as the electron shuttle to enhance iron cycling for more efficient hydroxyl radical production. Hence, we speculate that some products are responsible for such enhanced PhOH conversion in narrower nanopores, as clarified below.

We then quantified the products of PhOH oxidation in different systems. About 64 oxidation products were identified by IC and UHPLC-MS/MS (Supplementary Table 1 and Supplementary Figs. 11 and 12 for details), which can be divided into four categories based on the structure, i.e., quinones (BQ and HQ), aliphatic carbonyl compounds (aldehydes and ketones), organic acids, and oligomers (dimers, trimers, and tetramers). Over 85% of the products (in carbon

mass) were detected at 50% PhOH conversion in each catalytic oxidation system (Fig. 2b). About 2–50% of the products, most being organic acids, were adsorbed on the catalysts. The ratio of the adsorbed products varied along with the oxidation processes (Supplementary Fig. 13). The evolution of the yields of the products versus PhOH conversion was plotted in Fig. 2c–f. The yield of quinones, though gradually increasing, remained below 4.0% in all the oxidation systems since they could be continuously transformed into ring-opening molecules (vide infra). Narrowing the nanopores marginally increased the yield of quinones (Fig. 2c), a trend consistent with the enhanced PhOH conversion (Fig. 2a). Extra addition of 10 μM BQ or HQ into the catalytic oxidation systems showed that BQ, rather than HQ, was capable of shortening the lag phase of PhOH conversion (Supplementary Fig. 14). The reinforcing effect gradually diminished as the concentration of BQ decreased from 10 to 1.0 μM (Supplementary Fig. 15), implying the crucial role of BQ in accelerating PhOH conversion under spatial nanoconfinement. Opposite to the trend of quinone formation, the yields of ring-opening products significantly decreased in narrower pores (Fig. 2d, e). For example, the yield of organic acids was 17.9% at 50.0% PhOH conversion in bulk Mn$_3$O$_4$/PMS, and monotonically decreased to 2.7% in Mn$_3$O$_4$@20ACNT/PMS (Fig. 2d). The lower yields of ring-opening products suggest the suppression of degradation pathways in narrower nanopores.

In contrast to the suppressed degradation paths, the yield of oligomers (~36.7%) at 50.0% PhOH conversion in Mn$_3$O$_4$@20ACNT/PMS was 1.9, 2.3, and even 12.7 times that in Mn$_3$O$_4$@55ACNT/PMS (19.8%), Mn$_3$O$_4$@120ACNT/PMS (~16.1%), and bulk Mn$_3$O$_4$/PMS (~2.9%), respectively (Fig. 2f). The selectivity of oligomers in Mn$_3$O$_4$@20ACNT/PMS was higher than 60% at PhOH conversion of <60%, with the highest selectivity of 87.4% at 17.8% conversion. Comparably, the oligomer selectivity was much lower in Mn$_3$O$_4$/PMS (Supplementary Fig. 16). Moreover, the distribution of dimers, trimers, and tetramers at varied PhOH conversion (Fig. 2g) manifests a higher average degree of oligomerization (calculated as $\frac{\sum n \times \text{yield oligomers with n rings}}{\text{yield of oligomers}}$ ($n$ = 2, 3, and 4)) in narrower pores (Supplementary Fig. 17). As the reactions proceeded, the yield and selectivity of oligomers (Fig. 2f and Supplementary Fig. 16) decreased, which could be ascribed to either overoxidation of oligomers or formation of highly polymerized products adsorbed on the catalysts[9,10,40]. Overoxidation of oligomers occurred in all the oxidation systems as evidenced by the generation of degradation products bearing 7–11 carbon atoms (Supplementary Table 1 and Supplementary Fig. 18); however, the lower yields of the oligomer degradation products in Mn$_3$O$_4$@$n$ACNT/PMS than in Mn$_3$O$_4$/PMS contradict with the faster conversion of oligomers (cf. Fig. 2f and Supplementary Fig. 18). Meanwhile, careful analysis of the mass spectrometry data shows the absence of other degradation products from oligomer conversion. The conversion of oligomers to polymers on heterogeneous surfaces has been verified in recent studies[9,10,41]. Analysis of the products extracted from the catalysts (see Supplementary Method 1 for the procedure) by the matrix-assisted laser desorption/ionization time-of-flight mass spectrometry (MALDI-TOF-MS) and gel permeation chromatography confirmed the formation of highly polymerized products with an average molecular weight of ~1174 Da under 20-nm nanoconfinement (Fig. 2h and Supplementary Fig. 19). Shorter-chain polymers were determined in the pores larger than 20 nm (Fig. 2h and Supplementary Figs. 19 and 20), implying that the polymerization pathway is enhanced in narrower ACNTs.

Furthermore, we located Mn$_3$O$_4$ NPs of similar size on the outer surface of the 20-nm and 55-nm ACNTs (denoted as Mn$_3$O$_4$/20ACNT and Mn$_3$O$_4$/55ACNT, respectively; see Supplementary Method 2 for fabrication details and Supplementary Fig. 21a–d for characterization) and evaluated the oxidation of PhOH and the oligomer selectivity in Mn$_3$O$_4$/20ACNT/PMS and Mn$_3$O$_4$/55ACNT/PMS. The kinetics of PhOH oxidation in both systems was faster than that in Mn$_3$O$_4$/PMS, but was

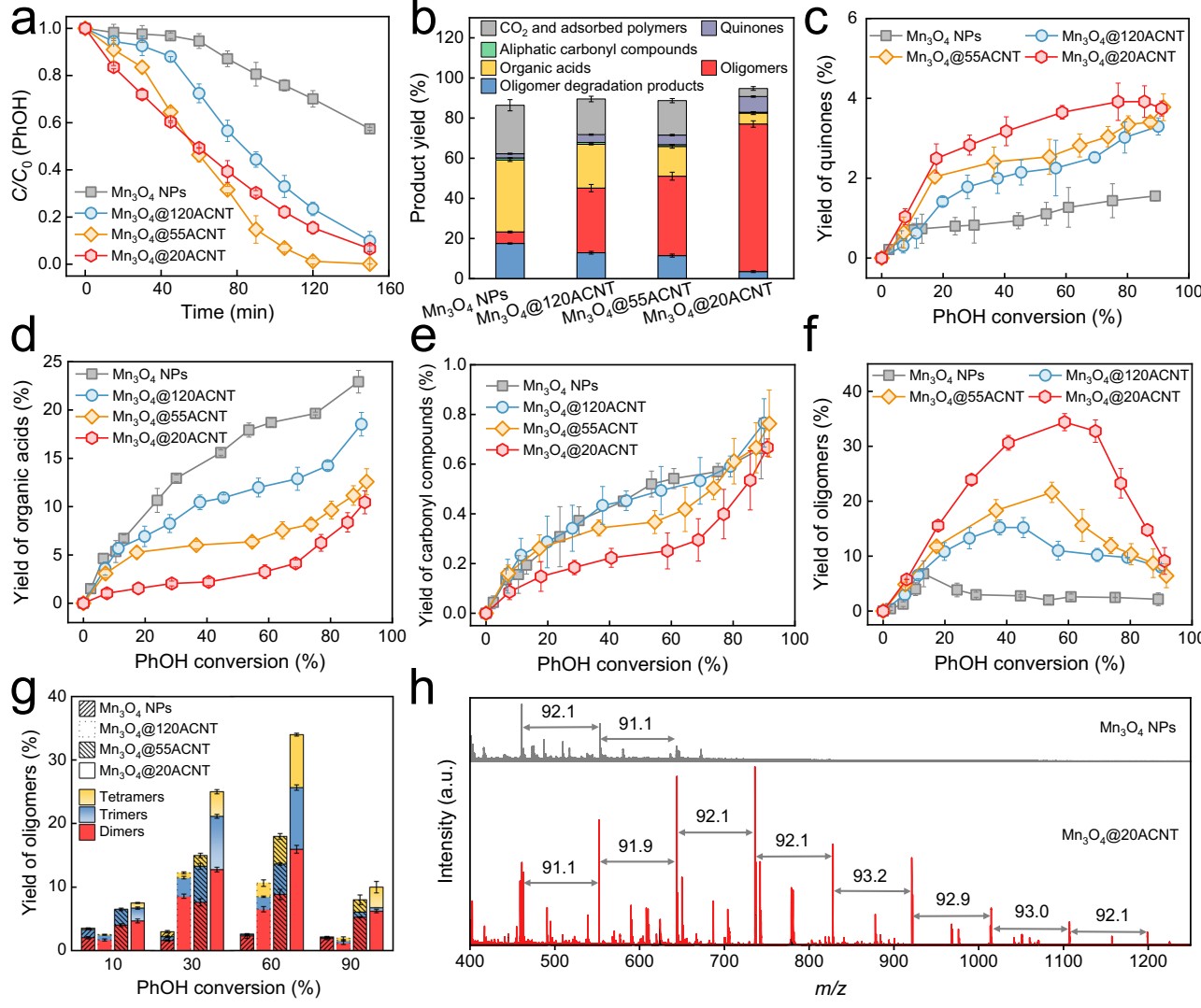

**Fig. 2 | PhOH conversion and product distribution in Mn₃O₄@*n*ACNT/PMS and bulk Mn₃O₄/PMS. a** Kinetics of PhOH conversion in different oxidation systems. **b** The yields of different products at 50.0% PhOH conversion. Plots of the yield of **c** quinones, **d** organic acids, **e** aliphatic carbonyl compounds, and **f** oligomers versus PhOH conversion in different oxidation systems. **g** Evolution of the yield of dimers, trimers, and tetramers during different oxidation processes. **h** MALDI-TOF-MS spectra of products adsorbed on the catalysts in Mn₃O₄/PMS and Mn₃O₄@20ACNT/PMS. The vertical scale is the same in the upper and lower chromatograms. Conditions: $T = 293.2 \pm 0.3$ K; pH = 7.0 ± 0.1; [catalyst] = 75 mg L⁻¹; [PhOH] = 200 µM; [PMS] = 2.0 mM. The error bars represent the standard deviations from triplicate tests.

significantly slower than that in the confined counterparts (Supplementary Fig. 21e, f). A similar trend for the yield of oligomers was also observed (Supplementary Fig. 21g, h), highlighting the crucial contribution of nanoconfinement to the enhanced oligomer selectivity.

It is well known that the degradation paths dominate contaminant removal in bulk oxidation systems, which produce complex and unidentified molecules with potential risks concomitant with the release of energy[42–45]. The preceding results foretell that redirection of carbon evolution pathways from degradation to polymerization, though difficult to achieve in bulk oxidation systems, can be feasibly realized by tuning the spatial size of nanoconfinement. The polymerization pathway consumes less PMS (Supplementary Fig. 22) while yielding products with higher internal chemical energy, which was roughly estimated by the enthalpy of reaction to be 5.98, 8.86, and 11.7 MJ mol⁻¹ for dimer, trimer, and tetramer, respectively (see Supplementary Table 2 and Supplementary Method 3 for detailed procedure). In addition, the degradation of PhOH and the yield of oligomers were reduced to varying degrees in all the oxidation systems during five continuous runs (see Supplementary Method 4 and Supplementary Fig. 23 for details),

possibly due to the accumulation of polymers on the surface, as further evidenced by the increased molecular weight of the polymers (cf. Supplementary Figs. 19 and 23g). It was also observed that the activity of the catalysts can be refreshed after washing off the polymers (see Supplementary Fig. 23 and Supplementary Method 1 for the procedure).

## Effect of solution chemistry on product selectivity

We are intrigued to know whether the enhanced oligomer selectivity under nanoconfinement holds in the aqueous environments of varying chemical properties. The decrease of PMS concentration from 2.0 mM to 200 µM retarded the degradation of PhOH and slightly reduced the yield of oligomers in all the catalytic oxidation systems (Supplementary Fig. 24). Lowering the pH values from 9.0 to 3.0 monotonically promoted PhOH conversion (Supplementary Fig. 25) and oligomer yield in all the systems (Fig. 3a–d). Since the concentration of neutral PhOH ($pK_a = 9.89$) approximately remains constant at pH < 7.0, the observed pH-dependence mainly arises from the change of the surface properties of Mn₃O₄ NPs ($pH_{pzc} = ∼4.5–4.8$; Supplementary Fig. 26). The presence of more protons favors the protonation of Mn₃O₄ NPs to

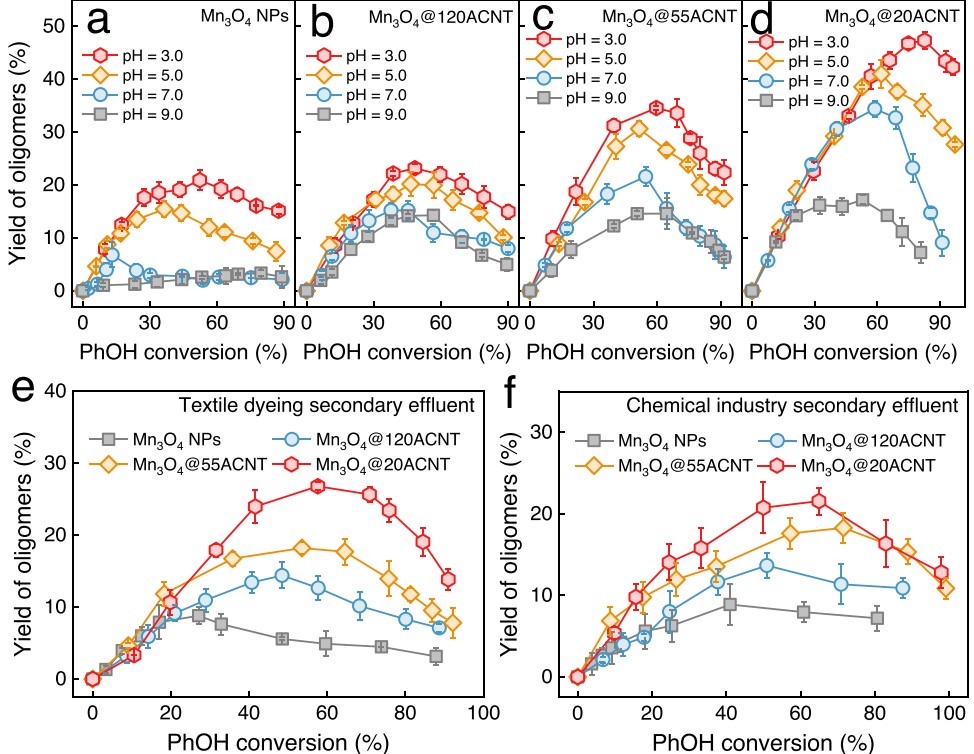

**Fig. 3 | Effect of solution chemistry on the yield of oligomers in Mn$_3$O$_4$@$n$ACNT/PMS and bulk Mn$_3$O$_4$/PMS.** Effect of pH on oligomer yield in **a** Mn$_3$O$_4$/PMS and **b**–**d** Mn$_3$O$_4$@$n$ACNT/PMS. Plots of the yield of oligomers versus PhOH conversion in the secondary effluent from **e** textile dyeing and **f** chemical industry wastewater treatment processes by different oxidation systems. Default conditions: $T = 293.2 \pm 0.3$ K; pH = $7.0 \pm 0.1$; [catalyst] = 75 mg L$^{-1}$; [PhOH] = 200 μM; [PMS] = 2.0 mM. The error bars represent the standard deviations from triplicate tests.

increase the reduction potential and/or to promote the formation of precursor complexes for efficient electron transfer[46,47]. Although Mn$_3$O$_4$ NPs hardly oxidized PhOH at pH 3.0–9.0 (Supplementary Fig. 27), the activity of Mn$_3$O$_4$ NPs toward PMS and/or the redox reactivity of in situ formed active Mn sites toward PhOH could be improved[47,48]. Furthermore, such improved oligomer selectivity in narrower pores can be observed in different industrial wastewaters (Fig. 3e, f and Supplementary Fig. 28a, b). The narrow pore-driven oligomer selectivity also holds for other aromatic contaminants including 4-chlorophenol, bisphenol A, and aniline (Supplementary Fig. 28c–h). The oligomer selectivity of these contaminants is highly dependent on their chemical reactivity toward active oxidants and the molecular structures, especially the available sites for coupling (Supplementary Fig. 29)[10,11]. These results underline the potential of nano-confinement in redirecting the pathways of carbon evolution in different water treatment scenarios.

## Identification of the pathways of PhOH conversion
The product distribution hinges on the nature of active oxidants and the subsequent oxidation processes. In heterogeneous peroxide-based AOPs, peroxides initially adsorb on the catalysts and cleave the catalyst–O or O–O bond to produce different active oxidants[9,49]. Previous studies have signified that nanoconfinement can steer the generation of active species that differ from the bulk reactions[34,50,51], thus altering the oxidation pathways and product distribution. Here, we identified the active oxidants and the elementary steps of PhOH conversion in these catalytic oxidation systems to shed light on the enhanced selectivity toward polymerization under nanoconfinement.

## The principal active oxidants
The produced active oxidants can either diffuse into the bulk solution or anchor onto the surface of catalysts. Spin trapping and chemical scavenging experiments were first conducted to evaluate the contribution of dissolved active oxidants. Figure 4a shows EPR spectra of the oxidation systems using 5,5-dimethyl-1-pyrroline $N$-oxide (DMPO) as the spin trap. Notwithstanding the presence of the signal of 5,5-dimethyl-1-pyrrolidone-2-oxyl (DMPOX), the signals of DMPO–HO$^\bullet$ and DMPO–SO$_4^{\bullet-}$ adducts were not observed. This is consistent with the negligible effect of the classical radical scavengers, i.e., methanol (MeOH) and $tert$-butanol, on the kinetics of PhOH conversion (Fig. 4b), indicating the absence of HO$^\bullet$ and SO$_4^{\bullet-}$ in all the systems. Using 2,2,6,6-tetramethyl-4-piperidone as the spin trap, the triplet signal of 2,2,6,6-tetramethyl-4-piperidinol-N-oxyl radical in the EPR spectra (Supplementary Fig. 30) indicates the existence of singlet oxygen ($^1$O$_2$). The lifetime of $^1$O$_2$ is notably shorter in water (2.9–4.6 μs) than in deuterium oxide (D$_2$O; 22–70 μs)[52]. As such, the rate of PhOH oxidation by $^1$O$_2$ is expected to increase in D$_2$O. However, the kinetic isotope effect (KIE) was not observed in all the concerned systems (Supplementary Fig. 31), excluding the potential role of $^1$O$_2$ in PhOH conversion.

More efforts were made to clarify the possible contribution of surface-bound active species, including the so-called Mn–PMS complexes and high-valent Mn species in the +IV or +V state. One can see that the catalysts pre-oxidized by PMS and washed by ultrapure water (Supplementary Method 5) were able to oxidize PhOH directly (Fig. 4c), implying the significant role of in situ generated surface-bound species in PhOH conversion. Moreover, the concentration of SO$_4^{2-}$ released (<3.7 μM, Supplementary Fig. 32a) was much lower than that of the oxidized PhOH (16.2–22.5 μM). Additional experiments confirmed that the catalysts did not adsorb SO$_4^{2-}$ (Supplementary Fig. 32b), evidencing the minor role of Mn–PMS complexes in PhOH conversion. Mn(V) is capable of oxidizing sulfoxides to sulfones via oxygen-atom transfer[53]. Using phenyl methyl sulfoxide (PMSO) as a chemical probe, the generation of phenyl methyl sulfone (PMSO$_2$) in

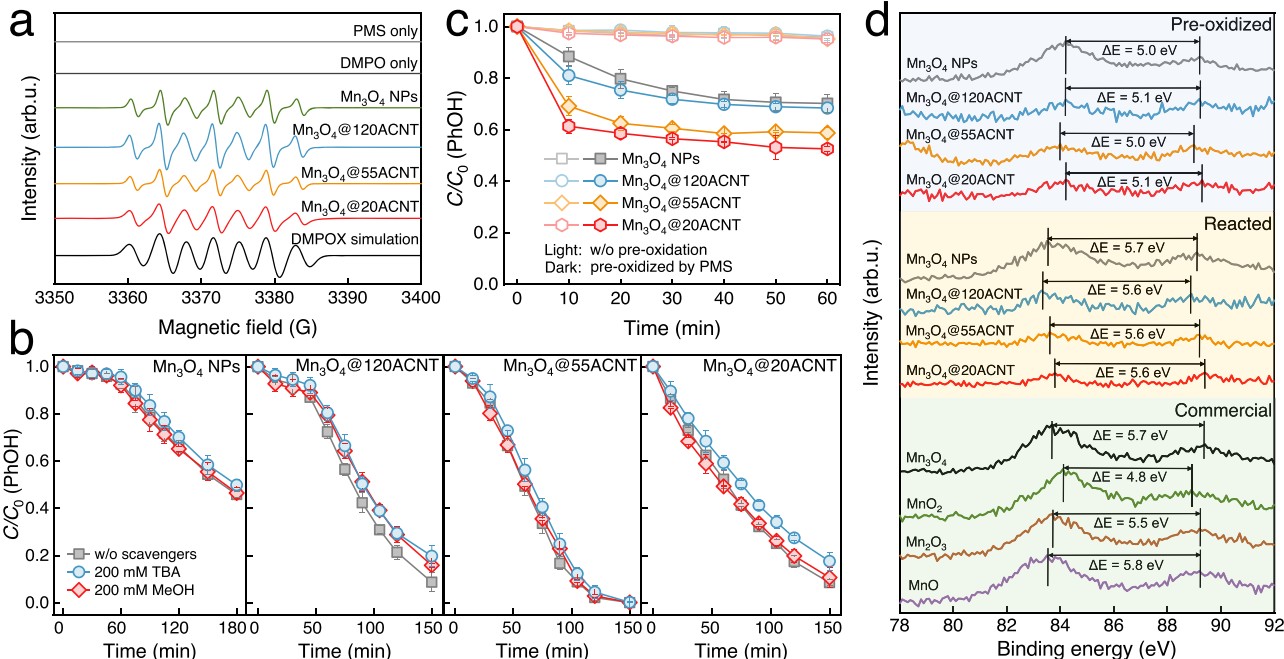

**Fig. 4 | Identification of the active oxidants. a** DMPO-trapped EPR spectra of different oxidation systems. DMPOX: $\alpha_N = 7.2$ G, $2\alpha_H^\gamma = 4.1$ G. **b** Effect of radical scavengers on PhOH conversion in $Mn_3O_4@nACNT$/PMS and $Mn_3O_4$/PMS. **c** Kinetics of PhOH oxidation by the pre-oxidized catalysts. **d** XPS spectra of Mn 3 s peaks of the catalysts pre-oxidized by PMS with some commercial catalysts for comparison. Default conditions: $T = 293.2 \pm 0.3$ K; pH = 7.0 ± 0.1; for **a**, **b** [catalyst] = 75 mg L$^{-1}$; [PhOH] = 200 μM; [PMS] = 2.0 mM; [DMPO] = 0/100 mM; for **c**, **d** [catalyst] = 400 mg L$^{-1}$; [PhOH] = 50 μM; [PMS] = 10 mM. The error bars represent the standard deviations from triplicate tests.

the catalytic oxidation systems was comparable to that in PMS alone (Supplementary Fig. 33), excluding the contribution of Mn(V). X-ray photoelectron spectroscopy (XPS) analysis of the catalysts pre-oxidized by PMS showed that the Mn 3 s multiplet splitting of $^7$S and $^5$S states decreased to ~5.1 eV[54,55], which was similar to that of the commercial $MnO_2$ (Fig. 4d). Further reaction of the pre-oxidized catalysts with PhOH restored the Mn 3 s multiplet splitting to ~5.6 eV (cf. Fig. 4d and Supplementary Fig. 4). Collectively, these results clarify that Mn(IV) was involved as the principal active oxidant for PhOH conversion in all the systems. Focusing on the active hydroxyl group of PhOH with a bond dissociation energy of ~88 kcal mol$^{-1}$ in $H_2O$[56], the KIE (i.e., $k_{PhOH}/k_{PDOD}$) was measured to be 1.0 ± 0.14 (Supplementary Fig. 34), demonstrating that conversion of PhOH proceeded via electron transfer.

## Elementary steps of PhOH conversion

Based on the above results, the elementary steps of PhOH conversion can be portrayed in Fig. 5a. The reaction between Mn(IV) and PhOH can proceed via either outer-sphere or inner-sphere electron transfer[47]. If the outer-sphere electron transfer occurred, direct abstraction of an electron from PhOH by Mn(IV) would give a phenol radical cation (PhOH$^{•+}$). PhOH$^{•+}$ has a p$K_a$ of −2.0 and a reduction potential of 1.50 V[57], and thereby either deprotonates rapidly (estimated to be >5.6 × 10$^5$ s$^{-1}$) to the phenoxy radical (PhO$^•$) or undergoes nucleophilic attack by $H_2O$ to form the dihydroxycyclohexadienyl radical[58]. At pH 7.0, most of the dihydroxycyclohexadienyl radicals should be oxidized by $O_2$ (~269 μM; >10$^9$ M$^{-1}$ s$^{-1}$) and the other oxidants (e.g., Mn(IV)) to HQ[59]. However, we found that reducing the $O_2$ concentration to ~25 μM by $N_2$ purging did not affect the product distribution in these systems (Supplementary Fig. 35). Additionally, NaNO$_3$ of 1.0 M, which is expected to mask the electrostatic interaction between the active Mn(IV) sites and PhOH, exhibited a negligible effect on PhOH oxidation by the pre-oxidized catalysts (Supplementary Fig. 36). These lines of evidence suggest the negligible role of the outer-sphere electron transfer process in PhOH conversion.

The inner-sphere electron transfer is slightly different from the outer-sphere electron transfer that PhOH oxidation by surface-bound Mn(IV) begins with the binding of PhOH on the surface to form a complex, followed by one-electron transfer to yield PhO$^•$ exclusively. The production of PhO$^•$ in these catalytic oxidation systems was verified by EPR analysis using DMPO as the spin trap (Fig. 5b, c). PhO$^•$ possesses an aromatic resonance structure with a higher spin density located at O, *para*-C, and *ortho*-C positions (see Supplementary Method 3 for theoretical computations), tending to undergo C−O or C−C coupling ($2k = \sim2.6 \times 10^9$ M$^{-1}$ s$^{-1}$)[60,61]. A close inspection of the dimer structure shows that more than 65% of the dimers generated in the initial stage originate from C−O and C−C coupling of PhO$^•$ in all the catalytic oxidation systems (Supplementary Fig. 37). Meanwhile, PhO$^•$ is not susceptible to $O_2$ attack ($k \le \sim1.0 \times 10^3$ M$^{-1}$ s$^{-1}$)[62], but might be oxidized to a phenoxenium cation (PhO$^+$) by surface-bound Mn(IV). The presence of PhO$^+$ in all the systems was supported by the formation of 4-chlorophenol from nucleophilic addition of Cl$^-$ to PhO$^+$[63] (Supplementary Fig. 38). Inasmuch as the positive charge of PhO$^+$ in the singlet ground state delocalizes into the aromatic ring, the electrophilic substitution of PhO$^+$ on PhOH leads to C−C coupling dimers preferentially[64]. Besides, hydrolysis and subsequent oxidation of PhO$^+$ produces quinones. Addition of Cl$^-$ reduced quinone formation (Supplementary Fig. 39a−d) and thus retarded the kinetics of PhOH conversion (Supplementary Fig. 39e−h). One-electron oxidation/reduction of the formed quinones gives semiquinone radicals, which could also undergo coupling to produce oligomers. However, the negligible formation of oligomers and the high selectivity toward organic acids during oxidation of quinones (Supplementary Figs. 40 and 41) indicate that further conversion of quinones mainly proceeds via degradation pathways. The above results manifest that PhO$^•$ serves as the crucial precursor in the polymerization pathway.

## Origin of enhanced oligomer selectivity under nanoconfinement

Nanoconfinement can trigger multiple effects, e.g., the host-guest interactions, the unique solution chemistry, and reactant enrichment,

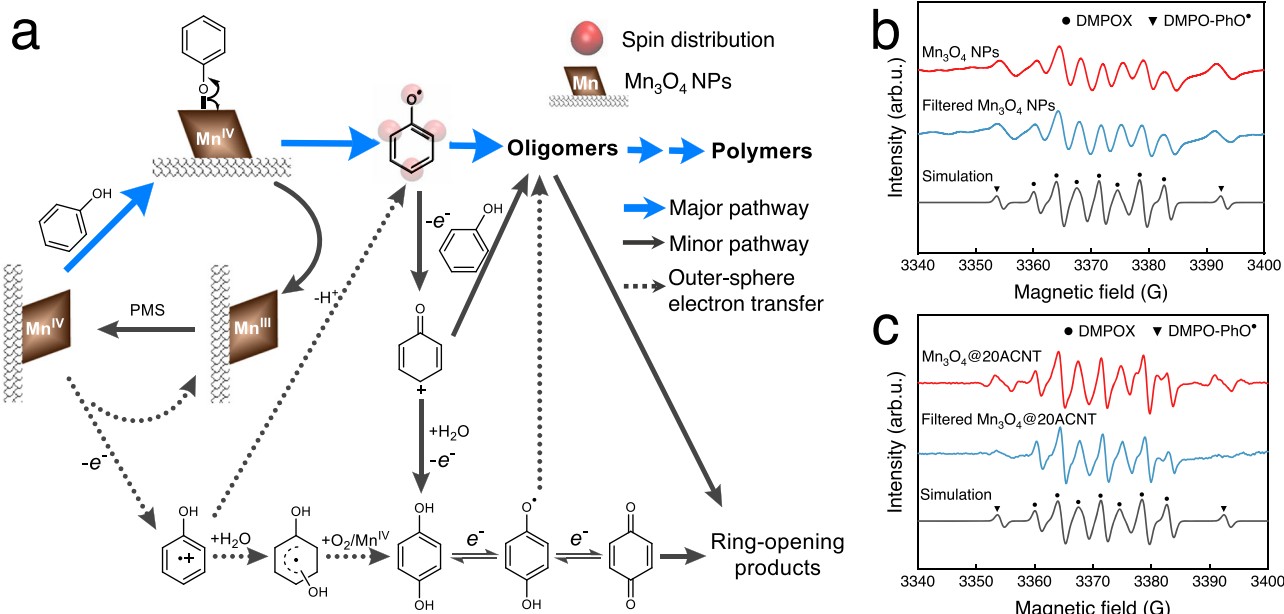

**Fig. 5 | The elementary steps of PhOH conversion. a** Scheme of the oligomerization and degradation pathways. The bold blue lines and solid gray lines indicate the major and minor paths of PhOH conversion, respectively. The dashed lines are the possible paths of the outer-sphere electron transfer process. EPR spectra of DMPO–PhO· adducts before and after filtration in **b** $Mn_3O_4$/PMS and **c** $Mn_3O_4$@20ACNT/PMS. DMPO–PhO· adducts: $\alpha_N = 13.1$ G, $\alpha_H^\beta = 9.1$ G. Conditions: $T = 293.2 \pm 0.3$ K; pH = $7.0 \pm 0.1$; [catalyst] = 300 mg L$^{-1}$; [PhOH] = 25 mM; [PMS] = 25 mM; [DMPO] = 100 mM.

to modulate the catalytic oxidation processes[35,65]. The above results unfold that the surface-bound Mn(IV) dominates PhOH conversion via single electron transfer regardless of whether $Mn_3O_4$ is confined. No experimental differences regarding the polymerization and degradation pathways were observed between the confined and bulk oxidation systems. Howbeit, the higher ratio of dimers from PhO· coupling (Supplementary Fig. 37) and more 4-chlorophenol generated from the addition of Cl$^-$ on PhO$^+$ (Supplementary Fig. 38) in narrower pores reasonably implies that the increase of PhO· concentration under nanoconfinement should be an important factor that favors PhO· self-coupling and/or cross-coupling with other intermediates (bold blue lines in Fig. 5a).

## The host-guest interactions

The cyclic voltammetry curves indicate that $Mn_3O_4$@20ACNT features the highest current density and greatest reductive capability, followed in sequence of $Mn_3O_4$@55ACNT, $Mn_3O_4$@120ACNT, and bulk $Mn_3O_4$ NPs (Fig. 6a). Moreover, the electrochemical impedance spectroscopy presents the better electric conductance of the confined catalysts (Fig. 6b). These results underpin that interactions of confined $Mn_3O_4$ NPs with ACNTs of smaller diameters promote the electron transfer processes for generation of active Mn(IV) sites and/or the subsequent PhOH conversion to PhO·.

## Lowering solution pH

Specific interactions of $H_2O$ with the catalyst surface and overlapping of the electrical double-layer under nanoconfinement drive the changes in water chemistry such as the solution pH[66]. The degree of pH variation is related to the structure of nanopores including the surface functional groups and pore size[67]. Since the surface of $Mn_3O_4$ NPs is of negative charge at pH 7.0, the concentration of protons is expected to be higher under nanoconfinement. The Donnan theory-based model was then employed to roughly calculate the average pH in the nanopores of varied sizes (see Supplementary Method 6 for detailed calculation). The average pH was estimated to be 3.7, 4.3, and 5.0 in the nanopores of 20, 55, and 120 nm in the background of neutral solution (Fig. 6c). The enrichment of protons near the surface of $Mn_3O_4$ NPs

confined in AAO nanopores was also visualized by molecular dynamic (MD) simulation in a recent study[68]. Lower pH is conducive to the polymerization pathway (Fig. 3a).

## Enrichment effect

The efficiency of polymerization would be significantly compromised by lowering the concentration of monomers[11,69]. A similar trend was observed in all the systems (Fig. 6d and Supplementary Fig. 42). In the presence of 10 μM PhOH, oligomers were barely formed in $Mn_3O_4$/PMS and $Mn_3O_4$@120ACNT/PMS. Nevertheless, one can still see the production of oligomers with a maximum yield of 18.0% and 10.0% in $Mn_3O_4$@20ACNT/PMS and $Mn_3O_4$@55ACNT/PMS at ~70.0% PhOH conversion, respectively. Narrower nanoconfinement is expected to enhance the enrichment of PhOH, intermediate radicals, and products. The increase of effective concentrations of PhOH and quinones favors PhO· generation. By virtue of EPR analysis, we found that most of PhO· were in the solution of $Mn_3O_4$/PMS, as evidenced by the nearly unchanged signal intensity of the DMPO–PhO· adducts following filtration. This is in contrast to the significantly decreased signal intensity in $Mn_3O_4$@20ACNT/PMS after filtration (Fig. 5b, c), suggesting the enrichment of PhO· inside the nanoconfined catalysts.

Furthermore, MD and finite-element method (FEM) simulations (see Supplementary Method 7 for details) were conducted to explore how the spatial size of nanoconfinement provided by ACNTs affects the reactions of phenoxy radicals. MD simulations with the ReaxFF reactive force field predict the increased collisions of PhO· molecules in narrower pores (Supplementary Fig. 43). In FEM simulations, hollow tubes (1.5 μm length, 7 nm wall thickness, and an inner diameter of 20, 55, 120, 1000 nm) with active interior surface were used to represent $Mn_3O_4$@nACNT and $Mn_3O_4$ NPs (Supplementary Fig. 44a–c). The chemical processes include adsorption and subsequent conversion of PhOH on the surface of catalysts, adsorption–desorption equilibrium of PhO·, coupling of PhO· to give dimers, and degradation of PhO· by the active surface to yield degradation products (Supplementary Fig. 44d–g). The mass transfer of the reactants and products is controlled by both bulk diffusion and Knudsen diffusion (see Supplementary Method 8 for calculation), with the effective diffusion

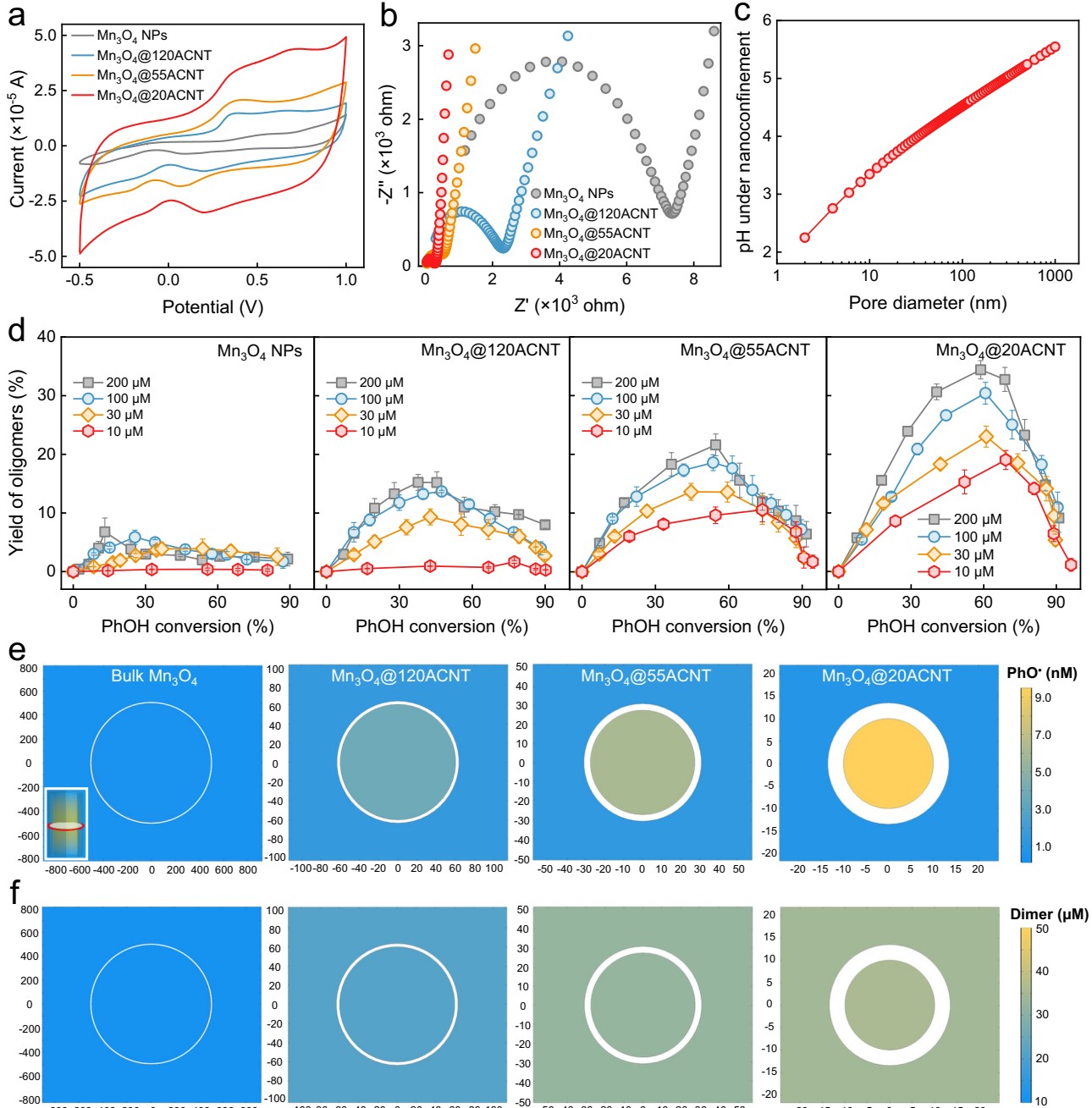

**Fig. 6 | The origins of product selectivity under spatial nanoconfinement. a** The cyclic voltammetry curves and **b** electrochemical impedance spectroscopy of different catalysts that were coated on the graphite electrode. **c** Modeling the pH in the nanopores in the background of a neutral solution. **d** Effect of PhOH concentration on the yield of oligomers in different oxidation systems. The computed concentration distribution of **e** PhO$^{\bullet}$ and **f** dimer in the mid-section of tubes in different oxidation systems at 50% PhOH conversion. The inset shows the 3D spatial distribution of PhO$^{\bullet}$ in the model and the position of the mid-section (red ring). The white circular ring represents the tube wall of 7.0 nm. The unit of the $x$- and $y$-axis is nm. Default conditions: $T = 293.2 \pm 0.3$ K; pH = $7.0 \pm 0.1$; [catalyst] = 75 mg L$^{-1}$; [PMS] = 2.0 mM; [PhOH] = 200 μM; for **a**, **b** scan rate = 2.0 mV s$^{-1}$, [Na$_2$SO$_4$] = 1.0 M. The error bars represent the standard deviations from triplicate tests.

coefficients detailed in Supplementary Table 3. We found PhO$^{\bullet}$ molecules were obviously restricted inside the tubes with smaller diameters (Fig. 6e). The concentration of PhO$^{\bullet}$ inside the 20-nm, 55-nm, 120-nm, and 1000-nm tubes was 10.5, 4.4, 2.3, and 1.0 times outside the tubes, respectively. A stronger enrichment effect is expected considering the reduced size of the available reaction zone due to the filling of Mn$_3$O$_4$ NPs. For instance, the size of nanoconfinement is approximately 5 nm in Mn$_3$O$_4$@20ACNT loaded with Mn$_3$O$_4$ NPs of 15.2 nm in diameter, resulting in a 35.3-fold higher PhO$^{\bullet}$ concentration in the pores than that in the bulk solution (Supplementary Fig. 45) and consequently more

effective collisions for further reactions (Supplementary Fig. 43). The selectivity toward polymerization was significantly enhanced under tighter nanoconfinement (Fig. 6f) and the ratio of polymerized products to fragmented products was satisfactorily correlated with the experimental results (Supplementary Fig. 46).

In summary, the carbon evolution pathways in the classical Mn$_3$O$_4$/PMS process can be readily tuned by varying the spatial size of nanoconfinement. Such a strategy is also extendable to different aromatic contaminants and water matrices. We believe that nanoconfinement holds the potential to regulate the product selectivity in a

range of heterogeneous oxidation systems except for the model $Mn_3O_4$/PMS system. Elaborate surface engineering of the supports to modulate the physicochemical property of the confined solution, the support-metal interactions, and reactant enrichment is expected to allow a rational elicitation of desired reaction outcomes.

Despite the above encouraging results, it should be noted that the proposed polymerization process under nanoconfinement is currently far from direct application in real wastewater treatment with energy recovery. Numerous efforts are still required to satisfactorily implement this new paradigm in real wastewater treatment scenarios. For example, the reaction selectivity toward polymers needs to be improved to achieve more efficient decontamination. Moreover, the produced polymers would either adsorb strongly on the catalysts or remain dissolved in the aqueous solution, challenging their facile separation from water as well as the reusability of the solid catalysts. Therefore, it is desirable to advance the nanoconfined catalytic oxidation systems to modulate the properties of formed polymers, and to develop compatible ancillary separation processes to facilitate polymer collection and thereby pave the way for subsequent energy recovery.

## Methods

### Materials and chemicals

The AAO membranes (60 μm thickness and 25 mm diameter) with different pore diameters were purchased from Hefei Pu-Yuan Nano Technology Co., Ltd. (China). PhOH and PMS were purchased from Sigma-Aldrich. Details of the product standards and other chemicals are presented in Supplementary Method 9. All chemicals were used as received and dissolved by ultrapure water (18.25 MΩ cm$^{-1}$). The actual wastewater samples were obtained from textile dyeing and chemical industry wastewater treatment processes (Supplementary Table 4 for detailed characterizations).

### Fabrication of the catalysts

The $Mn_3O_4$ NPs were confined in the ACNTs according to the scheme in Fig. 1a. Amorphous carbon was first introduced inside the AAO membranes by chemical vapor deposition (CVD). The CVD process proceeded at 1023 K for 120 min using ethylene as the carbon precursor (5% in Ar, 40 mL min$^{-1}$). The carbon-deposited AAO membranes were then subjected to hydrothermal reactions to confine $Mn_3O_4$ NPs inside the nanopores (see Supplementary Method 2 for details). The NPs possibly conglutinated outside the pores during fabrication were removed by ultrasonic washing with 1.0 mM $H_2SO_4$ and ethanol in sequence. Subsequently, the polished membranes were etched in 5.0 M NaOH solution at 323 K for 12 h to remove the AAO templates; the obtained materials of ~60 μm in length in the aqueous mixtures were then truncated by ball milling. Afterward, the mixtures were filtered, washed by ultrapure water, and vacuum-dried at 313 K for 18 h. The as-prepared catalysts were referred to as $Mn_3O_4@n$ACNT ($n = 20$, 55, and 120 representing the pore diameters of ACNT), and stored in a vacuum oven at 298 K. ACNTs were prepared following a similar process excluding the hydrothermal reactions to load $Mn_3O_4$ NPs. Conversely, bulk $Mn_3O_4$ NPs were synthesized via similar hydrothermal reaction and post-processing in the absence of the carbon-deposited AAO templates (Supplementary Method 2).

### Characterization of the catalysts

The crystalline structures of bulk and nanoconfined $Mn_3O_4$ were characterized by XRD (D8 Advance diffractometer, Bruker) using Cu Kα radiation (λ = 1.54056 Å) at 40 kV. The chemical states of the elements near the catalyst surface were determined by XPS (K-Alpha, Thermo Scientific). The morphologies were recorded on both TEM (JEM-2100 HR, JEOL) and scanning electron microscopy (SEM, S-3400 II, Hitachi). The EDX was performed to map the dispersion of the elements in the catalysts. The surface properties of the catalysts were examined by a LabRAM Aramis Raman spectrometer (Horiba

Scientific) with an excitation laser at 532 nm. The pore size distribution and specific surface area of the catalysts were measured using $N_2$ adsorption–desorption tests at 77 K (Autosorb iQ-MP, Quantachrome instruments). The loaded Mn contents of the catalysts were determined by inductively coupled plasma optical emission spectrometer (ICP-OES, iCAP7400, Thermo Scientific) after concentrated HCl-assisted digestion.

### Catalytic oxidation experiments

Chemical oxidation reactions were conducted in the glass beakers of 20 mL (for oxidation kinetics without product analysis) or 500 mL (for product analysis) at 293.2 ± 0.3 K under magnetic stirring. The reactors were wrapped with aluminum foils to avoid the possible influence of light. The pH value of the solution was adjusted to 7.0 ± 0.1 using 0.10 M $HNO_3$ and NaOH before the reaction and was barely changed during the oxidation processes (<0.3). The reaction was initiated by introducing 75 mg L$^{-1}$ catalyst into the mixture of 200 μM PhOH and 2.0 mM PMS. Reaction aliquots were withdrawn periodically and filtered through 0.22 μm polytetrafluoroethylene (PTFE) membranes to remove the catalysts. Note that the PTFE membranes do not adsorb the reactants and products. The filtrates were preserved with 0.50 M sulfite for analysis. Ultrahigh-performance liquid chromatography (UHPLC, UltiMate 3000, Thermo Scientific) equipped with a diode array detector was used to determine the concentration of residual organic compounds in solution (Supplementary Table 5 for detailed operational conditions). The concentration of PMS was measured by the KI method (Supplementary Method 10). Electron paramagnetic resonance (EPR) spectra were recorded using an ESR A300 spectrometer (EMX-10/12, Bruker) to detect the reactive oxygen species (ROS) and organic radicals (Supplementary Method 11).

### Analysis of the oxidation products

The scheme for product analysis is depicted in Supplementary Fig. 47. The products from PhOH oxidation can be either dissolved in solution or adsorbed on the catalyst surfaces. For analysis of the dissolved products, the reaction solution of 30 mL was sampled and filtered through 0.22 μm PTFE membranes. The filtrates were divided into three parts, named solution A, solution B, and solution C, respectively. Solution A containing 10 mL filtrate was quenched by 1.0 mL MeOH and was subjected to the UHPLC system coupled with a high-resolution Q Exactive Focus Orbitrap mass spectrometer (UHPLC-MS/MS, Thermo Scientific) with an electron spray ionization (ESI) source (Supplementary Table 6 for detailed operational conditions) to determine the relatively hydrophobic organic products, most bearing aromatic structures. The analysis was operated under negative ESI mode (Supplementary Method 12). Solution B containing 5.0 mL filtrate was quenched by 0.50 mL DMSO and derivatized with $p$-toluenesulfonyl hydrazine to analyze aldehydes and ketones on UHPLC-MS/MS under positive ESI mode (Supplementary Method 12). An integrated suspect and nontarget screening strategy was employed to identify the oxidation products on UHPLC-MS/MS (Supplementary Method 13). Solution C containing 10 mL filtrate was quenched by 1.0 mL 0.50 M nitrite and then was analyzed for organic acids by ion chromatography (IC; Supplementary Method 14). The concentrations of organic acids, HQ, and BQ were quantified using the available standard substances. The standards for most of the oligomers are not available commercially, and the concentrations were semi-quantified according to the prevailing protocols. For analysis of the products adsorbed on the catalysts, the PTFE membrane loading the filtered catalysts was transferred into a glass vial containing 9.0 mL 0.40 M HCl solution. The reactor was sealed and incubated at 313 K for 6 h to dissolve $Mn_3O_4$ NPs. The aqueous mixture was filtered and diluted to 30 mL with ultrapure water for analysis. No more oxidation products on the residual ACNTs after treatment can be extracted by organic solvents with varied polarity (Supplementary Method 15). Analysis of

the components in the as-obtained solutions follows a similar process for the dissolved products as described above. The sum of the generated $CO_2$ and the organic products strongly adsorbed on ACNTs (i.e., polymers) were quantified by the change in total organic carbon (TOC) before and after oxidation. Briefly, approximately 15 mL of the reaction solution was withdrawn at desired intervals and filtered via 0.22 μm PTFE membranes. Simultaneously, the corresponding catalyst particles were mixed with 9.0 mL of 0.40 M HCl solution at 313 K for 6 h to desorb the residual organic products. Subsequently, the filtrate and the desorption solution were subjected to a Shimadzu TOC-VCPH analyzer for TOC analysis, and the sum of both TOC values represents TOC after oxidation for the above calculation. The conversion, selectivity, and yield of the oxidation products are defined traditionally as shown in Supplementary Note 1.

## Data availability

The data that supports the findings of the study are included in the main text and supplementary information files. Raw data can be obtained from the corresponding author upon request. Source data are also provided as a Source Data file. Source data are provided with this paper.

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

## Acknowledgements

We sincerely thank Fangzhou Liu and Jiahe Zhang from the New Jersey Institute of Technology for their help on FEM simulations. This study was financially supported by National Natural Science Foundation of China (Grant No. 21925602/22236003/22106069), National Key R&D Program of China (2022YFA1205600), Natural Science Foundation of Jiangsu Province (Grant No. BK20210188), and the project funded by China Postdoctoral Science Foundation (2022M711557/2022T150309). We are also grateful to the High Performance Computing Center (HPCC) of Nanjing University for the numerical calculations on its blade cluster system.

## Author contributions

X.G., B.P., and Z.Y. conceived the research and designed experiments; B.P. supervised the progress of research; X.G. and Z.Y. conducted experiments; X.G., Z.Y., and B.P. discussed and analyzed data; W.Z. performed the FEM simulation; Z.Y. and X.G. drafted the manuscript; B.P. and W.Z. revised the manuscript; and X.G., B.P., Z.Y., and W.Z. commented on the manuscript.

## Competing interests

The authors declare no competing interest.
