## [Peer Review File · Nature Communications]

Carbon Redirection via Tunable Fenton-like Reactions Under Nanoconfinement Toward Sustainable Water TreatmentREVIEWER COMMENTS

Reviewer #1 (Remarks to the Author):

The chemical oxidation processes for water treatment have long been challenged by the out-of-control formation of various unknown products with potential risks. Gao et al present a very interesting and significant study attempting to tune the pathways of contaminant oxidation from fragmentation to polymerization via nanoconfinement. The redirection of carbon evolution pathways not only reduces the risk of undesired products of smaller molecular weights, but also provides a potential paradigm for energy harvesting during chemical oxidation. The manuscript is smooth and well-structured, and is quite extant in terms of experiments in different areas of interest. Overall, this study is thought provoking and certainly publishable in Nature Communications, provided that a revised manuscript could address some of the comments noted below.

Line 164, the authors suggested that more than 85% of products were determined if CO₂ is included. How was CO₂ quantified? That is, the concentration of CO₂ can be quantified directly as gas or as the difference in concentration of total organic carbon before and after reaction. If TOC variation before and after oxidation was used, the so-called CO₂ contains the contribution of polymers that are strongly adsorbed on the catalyst.

Line 207, how do the polymers adsorbed affect the catalytic activity of the catalysts? It is recommended to evaluate the reusability of the catalysts.

Line 232, the dosage of oxidant would affect the removal kinetics of contaminants and exert a discernible influence on product selectivity (Nat. Commun. 2022, 13, 3005; ACS ES&T Engg. 2021, 1, 1275). The results should be incorporated in this section.

Line 235, it is imperative to perform pH monitoring to ascertain if any substantial pH variations occur during the reaction process.

Line 250, please show more details of the coupling sites.

The authors present tubular models of 20 nm–120 nm in the simulations. However, since the particle size of Mn₃O₄ was ~15 nm, the spatial size of reaction zone should be smaller. A representative model with smaller spatial size should be simulated and analyzed to comprehensively exhibit the impact of nanoconfinement on reaction.

Figure 5 shows several pathways for phenol transformation. The authors are suggested to point out the major and/or minor pathways to be clearer for the readers.

Figure 6, the boundary of models is ambiguous and should be explained.

Reviewer #2 (Remarks to the Author):

The authors presented a nice study on carbon redirection via tunable Fenton-like reactions, for which nanoconfinement toward sustainable water treatment was claimed. A very comprehensive experimental design was presented, and a huge amount of data were used. The discussion was also clearly made. The paper is technically sound, however, the significance was overstated. Another major concern is that the conclusion was exclusive, while note being strongly supported by the data. Some details comments are below.

(1) Line 61-67: What concentration of phenol is used to do the energy calculation? When it is converted to polymers, another treatment such as adsorption, filtration, and/or flotation will be required. Then, what the energy input is required?

(2) It is said that the size of Mn_3O_4 is in the range of 15.0 – 24.0 nm, while the size of 20ACNT is about 19.8 nm. It is hard to understand a similar loading amount of 19.4wt% can be achieved. Also, Fig. 1 b,c,d clearly show that the density of particles inside the CNT is not even close.

(3) For supporting the claims, two sets of experiments might be helpful. a) Providing direct evidence to show all Mn_3O_4 were inserted into ACNTs, and b) Preparing three sized ANCTs supported Mn_3O_4 catalysts, which Mn_3O_4 are on the surface of ACNTs.

(4) The analysis of degradation pathways shows that a variety of intermediates, of course including oligomers, are produced. The recovery of oligomers from such a low concentration in the matrix of several tens of organics will consume much higher cost than it could possibly bring in.

(5) The conclusion was made on the hypothesis of that all ACNTs are same, all Mn_3O_4 is same, and all their properties are same. The exclusive effect of nanoconfinement was not strongly supported by the experimental/theoretical results.

Response to the comments from reviewers

Reviewer #1

1. *The chemical oxidation processes for water treatment have long been challenged by the out-of-control formation of various unknown products with potential risks. Gao et al present a very interesting and significant study attempting to tune the pathways of contaminant oxidation from fragmentation to polymerization via nanoconfinement. The redirection of carbon evolution pathways not only reduces the risk of undesired products of smaller molecular weights, but also provides a potential paradigm for energy harvesting during chemical oxidation. The manuscript is smooth and well-structured, and is quite extant in terms of experiments in different areas of interest. Overall, this study is thought provoking and certainly publishable in Nature Communications, provided that a revised manuscript could address some of the comments noted below.*

Response: Thanks for the positive evaluation in general. Now, we have carefully revised the manuscript to address all of your comments listed below.

2. *Line 164, the authors suggested that more than 85% of products were determined if CO₂ is included. How was CO₂ quantified? That is, the concentration of CO₂ can be quantified directly as gas or as the difference in concentration of total organic carbon before and after reaction. If TOC variation before and after oxidation was used, the so-called CO₂ contains the contribution of polymers that are strongly adsorbed on the catalyst.*

Response: Sorry for the possible ambiguous expression. In the original version we used the change of total organic carbon (TOC) before and after oxidation to indicate the generation of CO₂. As the reviewer suggested, the variation of TOC is in fact due to both the generated CO₂ and the products strongly adsorbed on ACNTs (i.e., polymers), which we fully agreed and the statement was revised. The method for TOC determination has been supplemented in the section of Methods in the revised manuscript (**Page 23, Lines 567–575**).

Page 7 Line 164: “Over 85% of the products (in carbon mass) were detected...”

Page 23 Line 567: “The sum of the generated CO₂ and the organic products strongly adsorbed on ACNTs (i.e., polymers) were quantified by the change in total organic carbon (TOC) before and after oxidation. Briefly, approximately 15 mL of the reaction solution was withdrawn at desired intervals and filtered via 0.22 μm PTFE membranes. Simultaneously, the corresponding catalyst particles were mixed with 9.0 mL of 0.40 M HCl solution at 313 K for 6 h to desorb the residual organic products. Subsequently, the filtrate and the desorption solution were subjected to a Shimadzu TOC-VCPH analyzer for TOC analysis, and the sum of both TOC values represents TOC after oxidation for the above calculation.”

3. *Line 207, how do the polymers adsorbed affect the catalytic activity of the catalysts? It is recommended to evaluate the reusability of the catalysts.*

Response: We appreciate the valuable comment. The reusability of the catalysts was evaluated to show more details about the effect of the adsorbed polymers on the kinetics of PhOH degradation as well as the product distribution. Results indicated that the accumulated polymers on the surface would, to varying contents, compromise the catalytic activity of the composite catalysts. Luckily, such negative effect will disappear after washing off the polymers from the catalyst (Fig. S23). Obviously, how to develop antifouling strategy of the confined catalyst is an important work to be concerned in future.

In the revised manuscript:

Page 10 Line 238: “In addition, the degradation of PhOH and the yield of oligomers were reduced to varying degrees in all the oxidation systems during five continuous runs (see Text S4 and Fig. S23 for details), possibly due to the accumulation of polymers on the surface, as further evidenced by the increased molecular weight of the polymers (*cf.* Fig. S19 and Fig. S23g). It was also observed that the catalytic activity of the catalysts can be refreshed after washing off the polymers (see Fig. S23 and Text S1 for the procedure).”

In the revised Supplementary Information:

Text S4. Evaluating the reusability of the catalysts.

Briefly, 0.02 g of the catalyst was mixed with 266.7 mL of 200 μ M PhOH. The reaction was initiated by adding 2.0 mM PMS. After the reaction, the catalysts were collected by suction filtration, and were lyophilized after rinsing with ultrapure water for three times. The resulting solids were subjected to the next run. To offset the loss of catalysts during operation, the volume of reaction solution was gradually decreased according to the mass of the obtained solids. After five continuous runs, the residual catalysts were extracted following the procedure detailed in Text S1. The extracted catalysts were subjected to another run following the same procedure described above.

Fig. S23. The reusability of the catalysts. The kinetics of PhOH degradation during five continuous runs and after regeneration in (a) Mn₃O₄/PMS and (b–d) Mn₃O₄@nACNT/PMS. Plots of the yield of oligomers versus PhOH conversion after (e) five continuous runs and (f) regeneration in different oxidation systems. (i) GPC spectra of the products extracted from Mn₃O₄ NPs and Mn₃O₄@nACNT after (g) five continuous runs and (h) regeneration. Conditions: T = 293.2±0.3 K; pH = 7.0±0.1; [PhOH] = 200 μM; [PMS] = 2.0 mM; [catalyst] = 75 mg L⁻¹.

4. Line 232, the dosage of oxidant would affect the removal kinetics of contaminants and exert a discernible influence on product selectivity (*Nat. Commun.* 2022, 13, 3005; *ACS ES&T Engg.* 2021, 1, 1275). The results should be incorporated in this section.

Response: Thanks for the reminder. We evaluated the effect of PMS concentration on the kinetics of PhOH oxidation and the generation of products. A drop of PMS concentration from 2.0 mM to 200 μM would retard PhOH oxidation and reduce the selectivity toward oligomers. The results have been supplemented in the revised manuscript and Supplementary Information.

In the revised manuscript:

Page 10 Line 248: “Decrease of PMS concentration from 2.0 mM to 200 μ M retarded the degradation of PhOH and slightly reduced the yield of oligomers in all the catalytic oxidation systems (Fig. S24).”

In the revised Supplementary Information:

Fig. S24. Effect of PMS concentration on (a) the kinetics of PhOH degradation and (b) the oligomer yield in different oxidation systems. Conditions: $T = 293.2 \pm 0.3$ K; $\text{pH} = 7.0 \pm 0.1$; $[\text{PhOH}] = 200 \mu\text{M}$; $[\text{catalyst}] = 75 \text{ mg L}^{-1}$.

5. Line 235, it is imperative to perform pH monitoring to ascertain if any substantial pH variations occur during the reaction process.

Response: The solution pH varied within 0.3 unit during the whole oxidation processes. Fig. R1 shows the evolution of pH in the bulk system and $\text{Mn}_3\text{O}_4@20\text{ACNT}/\text{PMS}$ system. The result has been supplemented in the revised manuscript.

Page 22 Line 529: “...and was barely changed during the oxidation processes (< 0.3).”

Fig. R1. Evolution of pH during degradation of PhOH in the bulk oxidation system and $\text{Mn}_3\text{O}_4@20\text{ACNT}/\text{PMS}$ system. Conditions: $T = 293.2 \pm 0.3$ K; $[\text{catalyst}] = 75 \text{ mg L}^{-1}$; $[\text{PhOH}] = 200 \mu\text{M}$; $[\text{PMS}] = 2.0 \text{ mM}$.

6. Line 250, please show more details of the coupling sites.

Response: We anticipated the coupling sites of the radicals generated from oxidation of bisphenol A, 4-chlorophenol, and aniline by DFT calculations and supplemented the results in the revised manuscript and Supplementary Information.

In the revised manuscript:

Page 11 Line 263: “The oligomer selectivity of these contaminants is highly dependent on their chemical reactivity toward active oxidants and the molecular structures, especially the available sites for coupling (Fig. S29)^{10,11}.”

In the revised Supplementary Information:

Fig. S29. Isosurface map of the electron spin density of the radicals generated from oxidation of bisphenol A, 4-chlorophenol, and aniline. The blue shadows indicate the high spin density. The unpaired electron is most likely located on the hydroxyl oxygen of bisphenol A and 4-chlorophenol, the amino nitrogen of aniline, and the *ortho* and *para* carbon atoms of the benzene ring. However, some of the sites, such as the *para* carbon of bisphenol A, seem unsuitable for the coupling reactions due to the steric effect.

7. The authors present tubular models of 20 nm–120 nm in the simulations. However, since the particle size of Mn_3O_4 was ~ 15 nm, the spatial size of reaction zone should be smaller. A representative model with smaller spatial size should be simulated and analyzed to comprehensively exhibit the impact of nanoconfinement on reaction.

Response: Thanks for the valuable comment. Using $Mn_3O_4@20ACNT$ as a representative, the effect of nanoconfinement with smaller spatial size on the reactions was investigated in the revised manuscript. MD simulation was used to study the collision of PhO^{\bullet} molecules within a 5-nm tube. In FEM simulations, a tube model with 5 nm inner diameter, 22 nm wall thickness, and 1.5 μm length was built and the spatial distribution of reactants and products was simulated. The results have been supplemented in the revised manuscript and Supplementary Information. In addition, the original FEM simulations simply assume a constant diffusion coefficient of PhO^{\bullet} in different tubes ($10^{-10} m^2 s^{-1}$). In the revised manuscript, we calculated the diffusion coefficients of the reactants in different tubes and updated the simulation results accordingly. The new results were similar to the original ones, demonstrating the enrichment of PhO^{\bullet} in the narrower tubes.

In the revised manuscript:

Page 19 Line 447: “The mass transfer of the reactants and products is controlled by both bulk diffusion and Knudsen diffusion (see Text S8 for calculation), with the effective diffusion coefficients detailed in Table S3.”

Page 19 Line 450: “The concentration of PhO[•] inside the 20-nm, 55-nm, 120-nm, and 1000-nm tubes was 10.5, 4.4, 2.3, and 1.0 times that outside the tubes, respectively. A stronger enrichment effect is expected considering the reduced size of the available reaction zone due to the filling of Mn₃O₄ NPs. For instance, the size of nanoconfinement is approximately 5 nm in Mn₃O₄@20ACNT loaded with Mn₃O₄ NPs of 15.2 nm in diameter, resulting in a 35.3-fold higher PhO[•] concentration in the pores than that in the bulk solution (Fig. S45) and consequently more effective collisions for further reactions (Fig. S43).”

Fig. 6. The computed concentration distribution of (e) PhO[•] and (f) dimer in the mid-section of tubes in different oxidation systems at 50% PhOH conversion. The inset shows 3D spatial distribution of PhO[•] in the model and the position of mid-section (yellow ring). The white circular ring represents the tube wall of 7.0 nm. The unit of the x and y axis is nm.

In the revised Supplementary Information:

Text S8. Diffusion of the reactants and products in the nanopores.

The mean free paths (λ) of the reactants and products (i.e, PhOH, PhO[•], dimer (C₁₂H₁₀O₂), and BQ) can be estimated according to the kinetic theory of gases.

$$\lambda = \frac{kT}{\sqrt{2}\pi d^2 p}$$

where k is the Boltzmann constant ($1.38 \times 10^{-23} \text{ J K}^{-1}$); d is the collision diameter of the target molecule, and p is the pressure. The Knudsen number (K_n) is calculated using the following equation to estimate the significance of Knudsen diffusion in the pores⁸.

$$K_n = \frac{\lambda}{d_{pore}} = \frac{kT}{\sqrt{2}\pi d^2 d_{pore} p}$$

The collision diameters of PhOH, PhO[•], C₁₂H₁₀O₂, and BQ are calculated to be 0.57 nm, 0.57 nm, 0.80 nm, and 0.57 nm, respectively, based on the optimized geometries (Text S3). Accordingly, K_n

of these compounds was calculated to be 0.12–1.38 in the pores of 20–120 nm (Table S3) and 0.014–0.028 in the pores of 1000 nm. Note that the mean free paths of the compounds would be much shorter in water than in gas. Assuming the mean free paths equal to the collision diameters of the concerned compounds, K_n in the pores of 20–55 nm (0.01–0.04) was also higher than 0.01, indicating the contribution of Knudsen diffusion to mass transfer. Lower K_n was obtained in the 120-nm and 1000-nm pores, suggesting the absence of Knudsen diffusion. For comparison, all the effective diffusion coefficients (D_{eff}) were calculated by the Bosanquet formula⁹.

$$\frac{1}{D_{eff}} = \frac{1}{D_{bulk}} + \frac{1}{D_{Knudsen}}$$

where D_{bulk} and $D_{Knudsen}$ are the coefficients of molecular diffusion and Knudsen diffusion, respectively. D_{bulk} was calculated according to the Wilke-Chang equation and $D_{Knudsen}$ was calculated by the following equation⁹.

$$D_{Knudsen} = \frac{d_{pore}}{3} \sqrt{\frac{8RT}{\pi M}}$$

where R is the gas constant, $8.314 \text{ J K}^{-1} \text{ mol}^{-1}$; M is the molar mass of the target. The values are detailed in Table S3.

Fig. S43. MD simulations of PhO'-involved reactions under nanoconfinement with different spatial sizes. (a) A snapshot of the established simulation models. (b) Effective collisions for PhO' coupling in different oxidation systems. The number of H₂O and PhO' were 2000 and 100, respectively. The temperature was set as 2000 K to facilitate collision within a limited interval.

Fig. S45. The computed concentration distribution of (e) PhO' and (f) dimer in the mid-section of a 5-nm tube at 50% PhOH conversion. The white circular ring represents the tube wall of 22 nm. The unit of the x and y axis is nm.

Table S3. The diffusion coefficients of the reactants and products.

	PhOH	PhO [•]	Dimer	BQ	
Collision diameter (nm)	0.57	0.57	0.8	0.57	
Molecular diffusion coefficient (m ² s ⁻¹)	7.87×10 ⁻¹⁰	8.02×10 ⁻¹⁰	5.29×10 ⁻¹⁰	7.87×10 ⁻¹⁰	
K_n	5 nm	5.53	5.53	2.81	5.53
	20 nm	1.38	1.38	0.70	1.38
	55 nm	0.50	0.50	0.26	0.50
	120 nm	0.23	0.23	0.12	0.23
	1000 nm	0.03	0.03	0.01	0.03
Kunsden diffusion coefficient (m ² s ⁻¹)	5 nm	4.28×10 ⁻¹⁰	4.30×10 ⁻¹⁰	3.04×10 ⁻¹⁰	3.99×10 ⁻¹⁰
	20 nm	1.72×10 ⁻⁹	1.72×10 ⁻⁹	1.72×10 ⁻⁹	1.72×10 ⁻⁹
	55 nm	4.71×10 ⁻⁹	4.73×10 ⁻⁹	3.35×10 ⁻⁹	4.39×10 ⁻⁹
	120 nm	1.03×10 ⁻⁸	1.03×10 ⁻⁸	7.30×10 ⁻⁹	9.58×10 ⁻⁹
	1000 nm	8.56×10 ⁻⁸	8.60×10 ⁻⁸	6.08×10 ⁻⁸	7.99×10 ⁻⁸
Effective diffusion coefficient (m ² s ⁻¹)	5 nm	2.77×10 ⁻¹⁰	2.8×10 ⁻¹⁰	1.93×10 ⁻¹⁰	2.65×10 ⁻¹⁰
	20 nm	5.4×10 ⁻¹⁰	5.47×10 ⁻¹⁰	4.05×10 ⁻¹⁰	5.4×10 ⁻¹⁰
	55 nm	6.85×10 ⁻¹⁰	6.86×10 ⁻¹⁰	6.47×10 ⁻¹⁰	6.67×10 ⁻¹⁰
	120 nm	7.44×10 ⁻¹⁰	7.44×10 ⁻¹⁰	7.23×10 ⁻¹⁰	7.27×10 ⁻¹⁰
	1000 nm	7.95×10 ⁻¹⁰	7.95×10 ⁻¹⁰	7.92×10 ⁻¹⁰	7.79×10 ⁻¹⁰

8. Figure 5 shows several pathways for phenol transformation. The authors are suggested to point out the major and/or minor pathways to be clearer for the readers.

Response: Thanks for the suggestion. Now we used bold blue lines to highlight the major pathways. We also explained the meaning of different arrows in the figure and the related caption. The figure has been updated in the revised manuscript.

Page 16 Line 390: “.....(bold blue lines in Fig. 5a).”

Fig. 5a. Scheme of the oligomerization and degradation pathways. The bold blue lines and solid grey lines indicate the major and minor paths of PhOH conversion in Mn₃O₄@nACNT/PMS, respectively. The dashed lines are the possible paths of the outer-sphere electron transfer process.

9. Figure 6, the boundary of models is ambiguous and should be explained.

Response: For better clarification, the white circular rings are here to represent the tube walls and has been stated in the caption of Figure 6 in the revised manuscript.

Fig. 6. The computed concentration distribution of (e) PhO* and (f) dimer in the mid-section of tubes in different oxidation systems at 50% PhOH conversion. The inset shows 3D spatial distribution of PhO* in the model and the position of mid-section (yellow ring). The white circular ring represents the tube wall of 7.0 nm. The unit of the x and y axis is nm.

Reviewer #2

1. *The authors presented a nice study on carbon redirection via tunable Fenton-like reactions, for which nanoconfinement toward sustainable water treatment was claimed. A very comprehensive experimental design was presented, and a huge amount of data were used. The discussion was also clearly made. The paper is technically sound; however, the significance was overstated. Another major concern is that the conclusion was exclusive, while not being strongly supported by the data. Some detailed comments are below.*

Response: We appreciate the thoughtful comments. As the reviewer suggested, we acknowledge that the polymerization process, while promising, is now just in the stage of infant and still far from direct application in wastewater treatment with energy recovery. Numerous efforts concerning the polymer selectivity, the separation and collection, and the appropriate pathways for waste-to-energy conversion are required in future to pave the way for possible application of this new paradigm in real wastewater treatment scenarios. In this study, we take a step to drive the transformation of the oxidation pathway from fragmentation to polymerization, which is hard to achieve in bulk oxidation systems, by regulating the spatial size of nanoconfinement, and to reveal the underlying mechanisms. In response to your comments, we have carefully evaluated and revised the related statements, and added further discussion on the above considerations to make the focus of this study clearer to the readers, as detailed in our response to Comments 2 and 5.

We realize that the effect of nanoconfinement was not fully supported by the results in the original manuscript, and have carefully revisited the lines of evidence that support our conclusion on the crucial role of spatial nanoconfinement in the polymer selectivity. While various credible characterizations confirm that these catalysts are of satisfactory similarity in structure, the ACNTs of varying sizes might impart distinct properties, resulting in the possible variations in the catalytic behaviors of different $\text{Mn}_3\text{O}_4@n\text{ACNT}$. To clarify this issue, we located Mn_3O_4 NPs of comparable size on the outer surface of the 20-nm and 55-nm ACNTs, and compared the oxidation product selectivity in $\text{Mn}_3\text{O}_4/20\text{ACNT}/\text{PMS}$ and $\text{Mn}_3\text{O}_4/55\text{ACNT}/\text{PMS}$ with that in the confined counterpart. The new results demonstrate the crucial role of nanoconfinement in the enhanced oligomer selectivity. The discussion has been supplemented in the revised manuscript as detailed in our response to Comment 6. The other comments were also carefully addressed as below.

2. *Line 61-67: What concentration of phenol is used to do the energy calculation? When it is converted to polymers, another treatment such as adsorption, filtration, and/or flotation will be required. Then, what the energy input is required?*

Response: Thanks for this valuable comment. For uniformity one mole of phenol is used for energy calculation. As calculated following the previous study¹, the internal chemical energy of the phenolic polymers is about 13.8 kJ g COD⁻¹. Assuming that the chemical oxygen demand (COD) of the phenolic wastewater is 500 mg L⁻¹ and all the phenolic compounds are transformed to polymers, the polymers formed from treating 1 m³ wastewater contains about 6.9 MJ of chemical energy.

Considering a 50% energy loss during energy transfer (i.e., from chemical energy to electrical energy by combustion), about 0.95 kWh can be produced theoretically. We scrutinized the energy consumption of filtration, air flotation, and adsorption in wastewater treatment plants. The values usually fall in the range of 0.005–0.03 kWh m⁻³ for filtration^{2,3}, 0.01–0.2 kWh m⁻³ for air flotation⁴, and 1.3–1.6 kWh m⁻³ for adsorption^{3,5,6}. While it seems that the recovered energy can offset such subsequent operations, it should be noted that these values would vary significantly according to the treatment capacity, the wastewater characteristics, the specific processes, and the operating conditions^{2,3,5}. Moreover, the energy consumption would remarkably increase by considering the other operating factors, for example, sludge dewatering and the production and regeneration of activated carbon for adsorption. Obviously, the proposed polymerization process is currently in the stage of scientific exploration instead of real application. Even though, in viewpoint of applied fundamental research, we believe that polymerization offers a promising route for sustainable wastewater treatment in conjunction with waste-to-energy conversion during the chemical oxidation processes, as exemplified by the pioneering works⁷⁻¹². For examples, the phenolic compounds in coal-conversion or coking wastewaters were polymerized by peroxidase or the heat/peroxymonosulfate process to precipitates that can be separated from water^{7,12}. In addition, the particular pollutant, polyvinyl alcohol, in the textile wastewater was polymerized to hydrochar with an energy yield of 118.5% by a hydrothermal reaction¹¹.

3. It is said that the size of Mn₃O₄ is in the range of 15.0 – 24.0 nm, while the size of 20ACNT is about 19.8 nm. It is hard to understand a similar loading amount of 19.4 wt% can be achieved. Also, Fig. 1 b, c, d clearly show that the density of particles inside the CNT is not even close.

Response: Thanks and your question may be explained by the following three aspects.

a. The confined catalysts, i.e., Mn₃O₄@nACNT, were prepared by the template method, with the procedure detailed in Methods in the manuscript and Text S2 in Supplementary Information. The key point of this procedure is the use of anodic aluminum oxide (AAO) membranes of varying diameters as the template. After Mn₃O₄ NPs were loaded, most of the particles conglomerated on the outer surface of the AAO membranes were removed by sequential ultrasonic washing with 1.0 mM H₂SO₄ and ethanol (Fig. R2). Subsequently, the AAO template was removed by alkali etching to obtain Mn₃O₄@nACNT. Therefore, no Mn₃O₄ NPs could be loaded on the outer surface of ACNT theoretically. The TEM images with a scale bar of 100 nm (Fig. 1b – d) and 0.5 μm (Fig. S3) clearly show the absence of nanoparticles outside the tubes.

Fig. R2. The acid-polished AAO membranes loaded with Mn₃O₄ NPs.

b. For comparison, we further made the new samples, i.e., Mn₃O₄ NPs loaded on the outer surface

of ACNTs (named as $\text{Mn}_3\text{O}_4/n\text{ACNT}$). TEM images of $\text{Mn}_3\text{O}_4/n\text{ACNT}$ clearly show that the locations of Mn_3O_4 NPs on $\text{Mn}_3\text{O}_4/n\text{ACNT}$ are significantly different from those on $\text{Mn}_3\text{O}_4@n\text{ACNT}$ (*cf.* Fig. S21a–b and Fig. 1c–d).

In the revised Supplementary Information:

Fig. S21. TEM images and EDX elemental mappings of (a) $\text{Mn}_3\text{O}_4/55\text{ACNT}$ and (b) $\text{Mn}_3\text{O}_4/20\text{ACNT}$.

c. We roughly calculated the density of particles inside the pores of different diameters. The volume of the tubes (Fig. R3) can be calculated by the following equation.

$$V = \pi(r_{\text{ex}}^2 - r_{\text{int}}^2)h$$

where r_{ex} and r_{int} represent the outer and inner radius of the tube, respectively. r_{int} is 20, 55, and 120 nm for $\text{Mn}_3\text{O}_4@20\text{ACNT}$, $\text{Mn}_3\text{O}_4@55\text{ACNT}$, and $\text{Mn}_3\text{O}_4@120\text{ACNT}$, respectively. r_{ex} is 7.0 nm larger than r_{int} , and h is the tube length (1.5 μm). It is reasonable to assume that ACNTs with different diameters have the same density. Therefore, the mass ratio of a single tube with r_{int} of 20, 55, and 120 nm is 3.29:8.19:17.29. Given the similar size of Mn_3O_4 NPs in different tubes, the number of Mn_3O_4 NPs confined in the 20-nm, 55-nm, and 120-nm tubes should be in a similar order (*i.e.*, 3.29:8.19:17.29) at a similar loading level (~ 20 wt%), resulting in the different density of particles within ACNTs of different sizes.

Fig. R3. A model of the tube material used in this study.

4. For supporting the claims, two sets of experiments might be helpful. a) Providing direct evidence to show all Mn_3O_4 were inserted into ACNTs, and b) Preparing three sized ACNTs supported Mn_3O_4 catalysts, which Mn_3O_4 are on the surface of ACNTs.

Response: Thanks for the suggestions. As seen from the response to Comment 3, we understand

that all the Mn_3O_4 NPs were inserted inside the tubes due to the Mn_3O_4 NPs loading inside the host prior to alkali etching of AAO membranes. We attempted to use 3D-TEM to provide direct evidence to show that all the Mn_3O_4 NPs were inserted inside the tubes. However, it proved challenging to make the tubes upright in the samples to be determined. Following the second suggestion, we loaded Mn_3O_4 NPs of similar size on the outer surface of ACNTs with a diameter of 20 nm and 55 nm, which are referred to as $\text{Mn}_3\text{O}_4/20\text{ACNT}$ and $\text{Mn}_3\text{O}_4/55\text{ACNT}$. $\text{Mn}_3\text{O}_4/120\text{ACNT}$ were not prepared because a lot of Mn_3O_4 NPs were inevitably loaded inside the tubes. The TEM images of $\text{Mn}_3\text{O}_4/20\text{ACNT}$ and $\text{Mn}_3\text{O}_4/55\text{ACNT}$ clearly show that the locations of Mn_3O_4 NPs are different from those of $\text{Mn}_3\text{O}_4@n\text{ACNT}$. Combined with the above analysis, it is believable that all the Mn_3O_4 NPs are confined within ACNTs. The TEM images of $\text{Mn}_3\text{O}_4/n\text{ACNTs}$ have been provided in the revised Supplementary Information.

In the revised Supplementary Information:

Fig. S21. TEM images and EDX elemental mappings of (a) $\text{Mn}_3\text{O}_4/55\text{ACNT}$ and (b) $\text{Mn}_3\text{O}_4/20\text{ACNT}$.

5. *The analysis of degradation pathways shows that a variety of intermediates, of course including oligomers, are produced. The recovery of oligomers from such a low concentration in the matrix of several tens of organics will consume much higher cost than it could possibly bring in.*

Response: Thanks very much for the valuable comment. We agree with the reviewer that recovery of oligomers from wastewater may require a variety of additional ancillary engineering techniques that could be more energy intensive. As seen in our response to Comment 2, we acknowledge that polymerization-based method for sustainable water treatment is still far from being applied in wastewater treatment, but, in viewpoint of applied fundamental research, it could be regarded as a potential strategy for sustainable wastewater treatment concurrent with waste-to-energy conversion during the chemical oxidation processes, as exemplified by some pioneering works (described in the response to Comment 2) ⁷⁻¹². While this study underscores the characteristics and origins of nanoconfinement-driven transformation of the oxidation pathway from degradation to polymerization, numerous efforts are still needed to improve the selectivity toward polymerization, to regulate the properties of polymers for easy separation and efficient energy recovery, to develop simple ancillary processes for polymer separation and collection, as well as to develop pathways for efficient waste-to-energy conversion in future to satisfactorily implement this paradigm. For better

clarity, we have carefully evaluated and revised the related statements, and added more discussion on the above considerations in the revised manuscript.

Page 3 Line 56: “Conceivably, one potential strategy for energy harvesting of dissolved organic contaminants is to convert them into polymers of higher molecular weights, which are usually of low solubility and could be separated from water by adsorption^{9,10}, filtration^{11,12}, or flotation¹³ for potential chemical refining¹⁴⁻¹⁷ and electricity production^{12,18,19}.”

Page 20 Line 469: “Despite the above encouraging results, the polymerization process under nanoconfinement is currently in the stage of infant, and still far from being applied in wastewater treatment with energy recovery. Numerous efforts are required to satisfactorily implement this new paradigm in real wastewater treatment scenarios. For example, the reaction selectivity toward polymers needs to be improved in order to achieve more efficient decontamination. Moreover, the produced polymers would either adsorb strongly on the catalysts or remain dissolved in the aqueous solution, challenging their facile separation from the water as well as the reusability of the solid catalysts. Therefore, it is desirable to advance the catalytic oxidation systems to modulate the properties of formed polymers and to develop compatible ancillary separation processes to facilitate polymer collection and thereby pave the way for subsequent energy recovery.”

6. *The conclusion was made on the hypothesis of that all ACNTs are same, all Mn₃O₄ is same, and all their properties are same. The exclusive effect of nanoconfinement was not strongly supported by the experimental/theoretical results.*

Response: Thanks for this valuable comment. Following the comment, we revisited the characterization of the catalysts. We confirm that all the ACNTs have similar wall thickness, similar amorphous wall structure, and also similar tube length (Figs. S1 and S2). Moreover, all the Mn₃O₄ NPs of similar size were confined in different ACNTs (Figs. 1a–d and S3). The content of Mn₃O₄ is also comparable in different Mn₃O₄@nACNT. While these catalysts are of satisfactory similarity in structure, the ACNTs with different sizes might feature distinct properties, resulting in the variations in the catalytic behaviors of different Mn₃O₄@nACNT. To clarify this issue, we located Mn₃O₄ NPs of similar size on the outer surface of the 20-nm and 55-nm ACNTs, and compared the product selectivity in the corresponding oxidation systems with that in the confined counterparts. Much lower oligomer selectivity was observed in Mn₃O₄/20ACNT/PMS and Mn₃O₄/55ACNT/PMS, clearly demonstrating the crucial role of nanoconfinement in the enhanced oligomer selectivity. We supplemented the related discussion in the revised manuscript.

Page 9 Line 220: “Furthermore, we located Mn₃O₄ NPs of similar size on the outer surface of the 20-nm and 55 nm ACNTs (denoted as Mn₃O₄/20ACNT and Mn₃O₄/55ACNT, respectively; see Text S2 for fabrication details and Fig. S21a–d for characterization) and evaluated the oxidation of PhOH and the oligomer selectivity in Mn₃O₄/20ACNT/PMS and Mn₃O₄/55ACNT/PMS. The kinetics of PhOH oxidation in both systems was faster than that in Mn₃O₄/PMS, but was significantly slower

than that in the confined counterparts (Fig. S21e–f). A similar trend for the yield of oligomers was also observed (Fig. S21g–h), highlighting the crucial contribution of nanoconfinement to the enhanced oligomer selectivity.”

Fig. S21. Characterization of Mn₃O₄/nACNT and PhOH conversion in Mn₃O₄/nACNT/PMS. TEM images and EDX elemental mappings of (a) Mn₃O₄/55ACNT and (b) Mn₃O₄/20ACNT. (c) Size-distribution histograms of Mn₃O₄ NPs in different catalysts. (d) XRD patterns of the catalysts. (e–f) The kinetics of PhOH degradation in Mn₃O₄/nACNT/PMS. (g–h) The yield of oligomers in Mn₃O₄/nACNT/PMS. The Mn contents of Mn₃O₄/20ACNT and Mn₃O₄/55ACNT were 15.5 wt% and 24.4 wt%, respectively. Conditions: T = 293.2±0.3 K; pH = 7.0±0.1; [PhOH] = 200 μM; [PMS] = 2.0 mM; [catalyst] = 75 mg L⁻¹.

Other revisions.

Some sentences were revised to improve the readability of the manuscript.

Page 3 Line 46: the phrase “($\sim 13\text{--}28 \text{ kJ g}^{-1}$ chemical oxygen demand; COD)” was revised to be “($\sim 13\text{--}28 \text{ kJ gCOD}^{-1}$; COD represents chemical oxygen demand)”.

Page 5 Line 116: the sentence “*The loaded Mn contents in the test composite samples were close to each other, i.e., 19.4 wt% for Mn₃O₄@20ACNT, 20.6 wt% for Mn₃O₄@55ACNT, and 21.3 wt% for Mn₃O₄@120ACNT.*” was revised to be “**The Mn contents in the three catalysts were very close as determined by acid digestion (i.e., 19.4 wt% for Mn₃O₄@20ACNT, 20.6 wt% for Mn₃O₄@55ACNT, and 21.3 wt% for Mn₃O₄@120ACNT).**”.

Page 5 Line 121: the sentence “*HRTEM images present that Mn₃O₄ NPs in all the catalysts exhibit an octahedral rhombus shape (Fig. 1e–h). The zoom-in images show (211) crystal lattice fringes (Fig. 1e–h).*” was revised to be “**HRTEM images indicate that Mn₃O₄ NPs in all the catalysts exhibit an octahedral rhombus shape with a dominant (211) crystal lattice plane (Fig. 1e–h).**”.

Page 7 Line 175: the sentence “*The reinforcing effect gradually diminished by decreasing the concentration of BQ from 10 to 1.0 μM (Fig. S15),...*” was revised to be “**The reinforcing effect gradually diminished as the concentration of BQ decreased from 10 to 1.0 μM (Fig. S15),...**”.

Page 10 Line 236: the sentence “*The polymerization pathway consumes less PMS while yielding products with higher internal chemical energy (Fig. S21 and Table S2). The heat that can be released from dimer, trimer, and tetramer is roughly estimated by the enthalpy of reaction to be 5.98, 8.85, and 11.72 MJ mol⁻¹, respectively (Table S2).*” was revised to be “**The polymerization pathway consumes less PMS (Fig. S22) while yielding products with higher internal chemical energy, which was roughly estimated by the enthalpy of reaction to be 5.98, 8.86, and 11.7 MJ mol⁻¹, respectively, for dimer, trimer, and tetramer (see Table S2 and Text S3 for detailed procedure).**”.

Page 12 Line 277: the sentence “*peroxides are initially adsorbed on the catalysts and the cleavage of the catalyst–O or O–O bond produces different active oxidants^{19,48}.*” was revised to be “**peroxides initially adsorb on the catalysts and cleave the catalyst–O or O–O bond to produce different active oxidants^{9,49}.**”.

Page 13 Line 315: the phrase “*the reference MnO₂*” was revised to be “**the commercial MnO₂**”.

Page 16 Line 382: the sentence “*The above results unfold that whether the Mn₃O₄/PMS process is confined, oxidation of Mn(II)/Mn(III) on Mn₃O₄ NPs by PMS produces the surface-bound Mn(IV), which dominates PhOH conversion via single electron transfer.*” was revised to be “**The above results unfold that the surface-bound Mn(IV) dominates PhOH conversion via single electron transfer, whether the Mn₃O₄/PMS process is confined.**”.

Page 24 Line 576: we added the statement of data availability. “The authors declare that the data supporting the findings of this study are available within the paper and the supplementary information. All data are available from the corresponding authors upon reasonable request.”

Page 28 Line 748: we added an acknowledgement: “We are also grateful to the High Performance Computing Center (HPCC) of Nanjing University for the numerical calculations on its blade cluster system.”

References

- 1 Heidrich, E., Curtis, T. & Dolfing, J. Determination of the internal chemical energy of wastewater. *Environ. Sci. Technol.* **45**, 827-832 (2011).
- 2 Longo, S. *et al.* Monitoring and diagnosis of energy consumption in wastewater treatment plants. A state of the art and proposals for improvement. *Appl. Energy* **179**, 1251-1268 (2016).
- 3 Mousel, D., Palmowski, L. & Pinnekamp, J. Energy demand for elimination of organic micropollutants in municipal wastewater treatment plants. *Sci. Total Environ.* **575**, 1139-1149 (2017).
- 4 Verstraete, W., Van de Caveye, P. & Diamantis, V. Maximum use of resources present in domestic “used water”. *Bioresour. Technol.* **100**, 5537-5545 (2009).
- 5 Bui, X., Vo, T., Ngo, H., Guo, W. & Nguyen, T. Multicriteria assessment of advanced treatment technologies for micropollutants removal at large-scale applications. *Sci. Total Environ.* **563**, 1050-1067 (2016).
- 6 Alhashimi, H. A. & Aktas, C. B. Life cycle environmental and economic performance of biochar compared with activated carbon: a meta-analysis. *Resources, Conservation and Recycling* **118**, 13-26 (2017).
- 7 Klibanov, A. M., Tu, T. M. & Scott, K. P. Peroxidase-catalyzed removal of phenols from coal-conversion waste waters. *Science* **221**, 259-261 (1983).
- 8 Zhang, X.-C. *et al.* Turning thiophene contaminant into polymers from wastewater by persulfate and CuO. *Chem. Eng. J.* **397**, 125351 (2020).
- 9 Wang, J. *et al.* Interlayer structure manipulation of iron oxychloride by potassium cation intercalation to steer H₂O₂ activation pathway. *J. Am. Chem. Soc.* **144**, 4294-4299 (2022).
- 10 Zhang, Y. *et al.* Simultaneous nanocatalytic surface activation of pollutants and oxidants for highly efficient water decontamination. *Nat. Commun.* **13**, 3005-3018 (2022).
- 11 Chen, C. *et al.* Innovative long-lasting catalytic hydrothermal reaction for high efficient energy harvest and carbon capture from recalcitrant wastewater. *Environ. Sci. Technol.* **57**, 11325-11335 (2023).
- 12 Li, J. *et al.* Abatement of aromatic contaminants from wastewater by a heat/persulfate process based on a polymerization mechanism. *Environ. Sci. Technol.*, doi.org/10.1021/acs.est.1022c06137 (2023).

REVIEWERS' COMMENTS

Reviewer #1 (Remarks to the Author):

The manuscript has answered the questions we asked and is recommended for acceptance.

Reviewer #2 (Remarks to the Author):

The authors carefully responded to the concerns from the reviewer, and provided new data to support the claims. Now the revised version is acceptable.